# Impaired Retromer Function in Niemann-Pick Type C Disease Is Dependent on Intracellular Cholesterol Accumulation

**DOI:** 10.3390/ijms222413256

**Published:** 2021-12-09

**Authors:** Kristina Dominko, Ana Rastija, Sandra Sobocanec, Lea Vidatic, Sarah Meglaj, Andrea Lovincic Babic, Birgit Hutter-Paier, Alessio-Vittorio Colombo, Stefan F. Lichtenthaler, Sabina Tahirovic, Silva Hecimovic

**Affiliations:** 1Laboratory for Neurodegenerative Disease Research, Division of Molecular Medicine, Ruder Boskovic Institute, 10000 Zagreb, Croatia; kristina.dominko@irb.hr (K.D.); Ana.Rastija@irb.hr (A.R.); lea.vidatic@irb.hr (L.V.); 2Laboratory for Mitochondrial Bioenergetics and Diabetes, Division of Molecular Medicine, Ruder Boskovic Institute, 10000 Zagreb, Croatia; ssoboc@irb.hr; 3Division of Biology, Faculty of Science, University of Zagreb, 10000 Zagreb, Croatia; sarahmeglaj@gmail.com (S.M.); andrea_babich@yahoo.com (A.L.B.); 4QPS Austria GmbH, 8074 Grambach, Austria; Birgit.Hutter-Paier@qps.com; 5German Center for Neurodegenerative Diseases (DZNE), 81377 Munich, Germany; Alessio-Vittorio.Colombo@dzne.de (A.-V.C.); Stefan.Lichtenthaler@dzne.de (S.F.L.); sabina.tahirovic@dzne.de (S.T.); 6Neuroproteomics, School of Medicine, Klinikum Rechts der Isar, Technical University of Munich, 81675 Munich, Germany; 7Munich Cluster for Systems Neurology (SyNergy), 81377 Munich, Germany

**Keywords:** cholesterol homeostasis, endolysosomal pathway, neurodegeneration, neurodegenerative diseases, NPC1, rare diseases, retromer

## Abstract

Niemann-Pick type C disease (NPC) is a rare inherited neurodegenerative disorder characterized by an accumulation of intracellular cholesterol within late endosomes and lysosomes due to NPC1 or NPC2 dysfunction. In this work, we tested the hypothesis that retromer impairment may be involved in the pathogenesis of NPC and may contribute to increased amyloidogenic processing of APP and enhanced BACE1-mediated proteolysis observed in NPC disease. Using *NPC1*-null cells, primary mouse NPC1-deficient neurons and NPC1-deficient mice (BALB/cNctr-*Npc1m1N*), we show that retromer function is impaired in NPC. This is manifested by altered transport of the retromer core components Vps26, Vps35 and/or retromer receptor sorLA and by retromer accumulation in neuronal processes, such as within axonal swellings. Changes in retromer distribution in NPC1 mouse brains were observed already at the presymptomatic stage (at 4-weeks of age), indicating that the retromer defect occurs early in the course of NPC disease and may contribute to downstream pathological processes. Furthermore, we show that cholesterol depletion in *NPC1*-null cells and in NPC1 mouse brains reverts retromer dysfunction, suggesting that retromer impairment in NPC is mechanistically dependent on cholesterol accumulation. Thus, we characterized retromer dysfunction in NPC and propose that the rescue of retromer impairment may represent a novel therapeutic approach against NPC.

## 1. Introduction

It is well established that changes in the endolysosomal system are linked to both the most common, as well as rare, inherited neurodegenerative diseases (NDs), indicating that maintaining endolysosomal homeostasis is important for brain cell function [1,2,3,4]. One of the key elements of the endosomal system is the retromer, a multimeric protein complex [5]. The retromer complex plays a critical role in maintaining endolysosomal function as it orchestrates cargo distribution [6,7] and recycling within the cell [8,9,10]. The center of the retromer is the cargo-recognition core that consists of proteins Vps26, Vps29 and Vps35 [6,7]. This trimer directly recognizes and binds to the cargo molecules (kinases, phosphatases and signaling receptors), forming a cargo-recognition complex. Intracellular sorting receptors, such as sorting-related receptors with A-type repeats (sorLA-, also known as SORL1 and LR11), acts as a bridge between the retromer and the cargo proteins and regulates their correct location and function within the cell. SorLA is expressed mainly in neurons in the CNS and is involved in the recognition of multiple cargo molecules and their retrograde trafficking from early endosomes to the trans-Golgi Network (TGN) via the retromer complex. Given that the retromer complex plays a critical role in endolysosomal trafficking and function, it is reasonable to suggest that the retromer and/or sorting receptor defect may be a central hub contributing to neurodegeneration.

Indeed, dysfunction of retromer has been associated with multiple NDs, including Alzheimer’s disease (AD), Parkinson’s disease (PD) [11] and recently, Amyotrophic Lateral Sclerosis (ALS) and Neuronal Ceroid Lipofuscinosis (NCL) [12,13]. There are both genetic and biochemical evidence that link retromer defect and neurodegeneration. In PD, a genetic mutation in Vps35 causes an autosomal dominant form of PD [14]. Multiple links between retromer defect and AD were reported. Genetic studies discovered variants in four genes coding for retromer or retromer-associated proteins that increase the risk of AD [15], as well as in a gene for the sorting receptor sorLA (*SORL1*) [16,17,18]. In addition, several mutations in *SORL1* cause a familial early-onset form of AD [19], underscoring the pathological contribution of retromer dysfunction in AD. Furthermore, AD patients of the most common “sporadic” late-onset form have lower levels of Vps26 and Vps35 proteins in the entorhinal cortex [20]. Retromer core components and sorLA regulate endosomal trafficking and proteolysis of the Alzheimer’s key protein β-amyloid precursor protein (APP) and, thus, formation of amyloid-β peptides (Aβ) that accumulate in AD patients’ brains [17,21,22]. Moreover, it has been shown that Vps35/retromer regulates the recycling of the microglial receptor TREM2 (triggering receptor expressed on myeloid cells 2), and *TREM2* rare variants were found to strongly increase the risk of developing AD, most likely by regulating microglial function, such as proinflammatory responses and phagocytosis [23,24,25]. Recent studies suggest that retromer dysfunction is also associated with tauopathies, such as tau hyperphosphorylation, aggregation and impaired cognition and behavior [26,27,28]. In ALS, a significant reduction of the retromer core components Vps35 and Vps26 was detected in motor neurons [12]. In addition, out of 13 NCL’s causative genes that affect endolysosomal function, at least 4 have been directly linked to retromers (*CLN1*, *CLN3*, *CLN5* and *CLN10* (cathepsin D, or CTSD)) [13]. Thus, it seems that retromer dysfunction is a shared feature among multiple neurodegenerative diseases and that retromer targeting could be a valuable common approach against NDs. Several retromer stabilizing agents, indeed, reduced key pathological features of AD and ALS, including Aβ and tau pathology, neuronal loss, memory and locomotor impairment, respectively [12,29,30,31].

It is intriguing that Alzheimer’s disease (AD) and a rare inherited lipid storage disorder Niemann-Pick type C (NPC) share several pathological features, including a dysfunctional endolysosomal system, enhanced amyloidogenic APP processing (resulting in accumulation of the C-terminal APP fragments (CTFβ/C99) and Aβ peptides), tau pathology, apolipoprotein E ε4 risk factor, activation of astrocytes and microglia (neuroinflammation), synaptic dysfunction and neurodegeneration [32,33]. As no effective therapy is available for NPC, most of the patients die between 10 and 25 years of age [34]. NPC, therefore, despite its relative infrequency (1:100,000 live births), imposes an emotional and economic burden on patients, families and society [34]. NPC is caused by mutations in *NPC1* (95% of cases) or *NPC2* (5% of cases) genes that code for late endosomal/lysosomal cholesterol trafficking proteins. NPC1 and NPC2 presumably act in concert in the transport of endocytosed low-density lipoprotein (LDL)-derived free (unesterified) cholesterol from late endosomes/lysosomes to other cellular compartments, such as the endoplasmic reticulum, TGN and plasma membrane. This process is impaired in NPC, resulting in the accumulation of free cholesterol and other lipids (gangliosides and lysobisphosphatidic acid—LBPA) in late endosomes/lysosomes causing endolysosomal dysfunction [35,36,37]. Although free cholesterol accumulation has been considered as a primary event in NPC, there are studies that suggest that NPC features are caused by defects in other lipids or even that cholesterol accumulation may be a secondary event [38]. Therefore, one needs to identify an offending lipid in NPC that initiates pathological processes and correlate it with a particular functional defect in order to employ an effective lipid-lowering strategy against NPC. Indeed, our recent work on the characterization of the NPC microglia showed that altered trafficking and cholesterol dyshomeostasis appear as major culprits of NPC pathology [36]. This has been supported by the preclinical and clinical trials with a cholesterol extracting compound cyclodextrin (2-hydroxypropyl-β-cyclodextrin, HPβCD) [39,40,41,42,43]. Indeed, the administration of HPβCD, both systematically and directly into the CNS, has shown the greatest disease amelioration in NPC1 mouse and feline models, including the rescue of lethality and improved pathological hallmarks of NPC disease [39,40]. However, ototoxicity, e.g., further impairment of hearing that is already present in NPC, was detected as the most severe adverse event. Importantly, direct intracerebroventricular (ICV) injection overcame HPβCD inability to efficiently cross the blood–brain barrier and resulted in complete prevention of neurodegeneration of Purkinje cells in NPC1 mice. Clinical trials with intrathecal (IT) HPβCD administration showed both biomarker and clinical evidence of efficacy in patients with NPC disease [41,42]. Although the doses applied were generally well-tolerated, dose-limiting side-effects, including the transient post-dose ataxia and fatigue, were reported, in addition to ototoxicity. Recent studies on intravenous (IV) administration of HPβCD, with or without IT application, to moderately and severely affected NPC patients showed neurologic and neurocognitive benefits in most patients with IV application alone, independent of the IT administration [43]. In some individuals receiving IV HPβCD, the NPC disease seemed less progressive. No unexpected safety issues were experienced, and the systemic reactions that were noted did not lead to treatment discontinuation. Notably, hearing loss was not reported in the patients receiving IV therapy, as has been reported in previous clinical trials on NPC patients with IT administration [41,42,43]. Thus, HPβCD treatment seems to play an important role in targeting the cellular cholesterol burden in NPC, but optimal dosing, dosing interval and route that would provide benefits for patients with NPC are still to be determined. It is likely that a multi-system disease, such as NPC, would require multitargeted therapy. In addition, the heterogeneity of the disease must be considered for treatment strategies against NPC, as an approach in one patient may not be the most appropriate for another.

Despite the number of similarities between AD and NPC, these disorders also show clear neuropathological differences. While in AD, neurodegeneration primarily affects the hippocampal region and cortex, and the cerebellum seems the least affected, in NPC, the loss of cerebellar Purkinje neurons represents the primary feature, followed by the loss of cortical neurons, while the hippocampus seems to be less affected by neurodegeneration [44,45]. Nevertheless, despite different regional susceptibility to neurodegeneration, we postulate that these two diseases may share common molecular pathway/s leading to neuronal dysfunction. While retromer dysfunction is clearly an important feature of AD, and AD and NPC show common and overlapping pathological features, in this work, we tested the role of targeting retromer in NPC as a possible novel therapeutic approach that we are desperately missing in NPC.

To elucidate the role of retromer in the pathogenesis of NPC disease, in this work, we characterized the levels and the localization/distribution of retromer proteins (Vps26 and Vps35) and retromer receptor sorLA in *NPC1*-null cells and in different brain regions of NPC1 mice (cerebellum, cortex and hippocampus) at different disease stages (4, 7 and/or 10-week old). Since retromer has an important role in synaptic function, we also analyzed retromer (dys)function in mouse primary hippocampal and cortical NPC1 vs. wt neurons, and, more specifically, in axonal swellings, which are one of the first hallmark(s) of neuronal dysfunction and may predict their vulnerability to degeneration. Finally, by modulating cholesterol levels (cholesterol depletion/loading) we investigated the causal link between altered cholesterol levels and retromer (dys)function. Here, we show that altered retromer trafficking/distribution is a characteristic feature of NPC disease that is dependent on cholesterol accumulation. Retromer defect was detected in all brain regions analyzed in NPC1-deficient mice, as well as in NPC1-deficient hippocampal and cortical neurons, supporting a general feature of retromer dysfunction triggered by the NPC1-loss and subsequent cholesterol accumulation. Whether rescue of retromer function per se may revert NPC pathological features still needs to be determined. However, our findings support that the accumulation of free cholesterol is likely the offending lipid in NPC disease and that cholesterol-lowering strategies may result in a specific targeting of altered trafficking of retromer and of the endolysosomal system leading to an effective therapy against NPC.

## 2. Result

### 2.1. Retromer Trafficking and Distribution Is Impaired in CHO NPC1-Null Cells

In our previous work we showed an increased processing of APP and of additional substrates, Sez6 and Sez6L, by a key Alzheimer’s protease, BACE1, in a cellular model of NPC disease (CHO cells with deleted *NPC1* gene, *NPC1*-null cells) and in NPC1-mouse brain tissue, respectively [46,47,48]. Since in AD, enhanced amyloidogenic processing of APP by BACE1 involves the dysfunction of retromer complex and, thus, sequestration of BACE1 and its substrate APP in early endosomal compartments [49,50], where acidic pH enables optimal cleavage by BACE1, we hypothesized that dysfunctional retromer transport causes enhanced BACE1-mediated proteolysis in NPC as well. To test the retromer function in NPC, we used CHO *NPC1*-null vs. CHOwt cells as a cellular model and monitored intracellular localization and protein levels of retromer core components Vps26 and Vps35 (Figure 1A). In *NPC1*-null cells, retromer proteins accumulated in enlarged peripheral vesicles in contrast to CHOwt, suggesting an altered trafficking and distribution of the retromer core complex (Figure 1A). Quantification of the immunofluorescent signal revealed that the size of the Vps26 and Vps35-puncta are significantly larger in *NPC1*-null compared to wt-cells (Figure 1B) and confirmed similar colocalization of the two retromer proteins in *NPC1*-null and CHOwt cells (Figure 1A,C). The analysis of protein levels of Vps26 and Vps35, as well as of the retromer receptor sorLA, did not reveal any differences between CHOwt and *NPC1*-null cells (Figure 1D,E), indicating that retromer trafficking dysfunction in a cellular model of NPC is not triggered by altered expression of its core components.

Since we observed altered retromer subcellular immunoreactivity in *NPC1*-null vs. CHOwt cells, we further analyzed the identity of accumulating Vps35-positive vesicles in *NPC1*-null cells. We co-immunostained Vps35-positive vesicles with different subcellular markers: TGN46 (trans-Golgi Network protein 2) for Golgi apparatus, EEA1 (early endosome antigen 1) for early endosomes, TfR (transferrin receptor) for recycling endosomes, Rab7 for late endosomes, LAMP1 (lysosomal associated membrane protein 1) for lysosomes and filipin staining for intracellular free cholesterol. *NPC1*-null cells showed altered immunostaining of all subcellular markers analyzed together with free cholesterol accumulation within late endosomes and lysosomes (Figure 2A–C, Appendix A). The observed enlarged early endosomes (Figure 2A,D), as well as enlarged recycling endocytic compartments (Figure 2B,D), in *NPC1*-null vs. CHOwt cells suggest that defects within the endolysosomal pathway in *NPC1*-null cells also involve endocytic organelles that are not the primary site of cholesterol accumulation due to NPC1/2 loss. Furthermore, in contrast to CHOwt cells, in *NPC1*-null cells, Vps35 accumulated in the enlarged TfR-positive (Figure 2B,E) and LAMP1-positive vesicles (Figure 2C,E). Although colocalization of Vps35 with early endosome marker EEA1 was found similar between CHOwt and *NPC1*-null cell (Figure 2A,E), in *NPC1*-null cells, Vps35 was co-stained in enlarged EEA1-positive vesicles, indicating a defect within the early endocytic organelles as well. While no colocalization of Vps35 and TGN46 was observed, there were few enlarged Vps35 and Rab7-positive vesicles in *NPC1*-null vs. CHOwt cells (Appendix A). In sum, these findings indicate that retromer trafficking is altered in *NPC1*-null cells and that retromers are mainly trapped within enlarged early/recycling endosomes and lysosomes, suggesting impaired retromer function. However, the altered trafficking of retromer components in *NPC1*-null cells was not due to altered levels of retromer and/or endolysosomal proteins since the levels of retromer proteins (Figure 1D,E), and of EEA1, TfR and LAMP1 (Appendix A) were comparable between CHOwt and *NPC1*-null cells.

To further characterize whether the altered distribution of retromer components in *NPC1*-null cells has functional consequences, we analyzed the colocalization of Vps35 and APP (Figure 3), a key Alzheimer’s protein that is transported out of early endosomes via retromers—a process that protects against Aβ formation and Alzheimer’s disease. We used a C-terminal APP antibody (APP-CT), which detects the full-length-APP protein (fl-APP) and the membrane-bound C-terminal APP fragments [46,47]. Staining with APP-CT antibody revealed distinct APP staining and increased colocalization with Vps35 in *NPC1*-null vs. CHOwt cells (Figure 3). Thus, our immunocytochemistry findings bring evidence that the lipid storage disorder NPC is characterized, not only by dysfunction at the stage of late endosomes/lysosomes, but also by alterations in earlier compartments, such as early endosomes and retromers, strengthening the importance of the trafficking defect in the pathophysiology of NPC disease.

### 2.2. Retromer Distribution Is Altered in NPC1-Mouse Brains

Due to a profound CNS involvement in the pathogenesis of NPC disease, we further analyzed if the defects in retromer localization observed in a cellular model of NPC could be translated into the brains of NPC1-deficient mouse model. To test this, we used the *NPC1*^−/−^ mice (BALB/cNctr-*Npc1m1N*, also known as *NPC1^nih^*, NPC1) that is the most widely-used mouse model, which recapitulates pathology of NPC disease in humans [51]. We analyzed retromer proteins in three differentially affected brain regions in NPC1 mice, namely, cerebellum, which shows a characteristic loss of Purkinje neurons, hippocampus, which is considered preserved and cortex, which also demonstrates degeneration of neurons, albeit at later stages. In addition, the brains were collected at different ages: from 4, 7 and 10-week old NPC1 mice that span from the early (presymptomatic) to end-stage of NPC disease. Since our results in *NPC1*-null cells revealed altered retromer trafficking, we firstly analyzed subcellular localization and distribution of retromer proteins in different neuronal types across the cerebellum, hippocampus and cortex of NPC1 vs. wt mice. Namely, we analyzed the brain distribution of retromer proteins Vps35 and sorLA. The staining of Vps35 and sorLA in 4, 7 and 10-week old wt mouse brains did not differ in either of the regions analyzed, and thus, the results of immunostaining of 10-week old wt mouse brains were selected as a representative control (Figure 4).

Our immunohistochemistry results revealed that Vps35 and sorLA immunoreactivity was distributed in the soma of Purkinje neurons and in the molecular layer of the cerebellum of wt mice (Figure 4A). In NPC1 mouse cerebellum, the immunostaining of both Vps35 and sorLA was altered compared to wt brains. SorLA immunostaining was significantly decreased in the soma of NPC1 Purkinje neurons, and this effect was already observed early in the pathogenesis of NPC disease in a presymptomatic stage at 4-weeks of age (Figure 4A,D). Furthermore, in parallel to sorLA decrease, we observed significant accumulation of the retromer protein Vps35 as small puncta in the soma of Purkinje neurons of NPC1 mouse cerebella already at 4-weeks of age. The accumulation of Vps35 further progressed with the pathogenesis of NPC disease and became more apparent at 7-weeks and especially in surviving Purkinje neurons of 10-weeks of age in NPC1 mice, in contrast to its diffuse signal in wt neurons (Figure 4A,D). Thus, the altered distribution of retromer proteins sorLA and Vps35 in NPC1 Purkinje neurons is an early event in the course of NPC disease that occurs already at the presymptomatic stage.

Immunostaining of sorLA and Vps35 was further analyzed in the hippocampus, which is considered protected against neurodegeneration in NPC disease, and also in the cortex, which shows degeneration of neurons in NPC at the later stage. We further monitored whether altered retromer distribution is a general feature of an endolysosomal defect in NPC1 mouse brains and/or occurs only in neurons more vulnerable to neurodegeneration, such as Purkinje neurons. Indeed, in the hippocampus (Figure 4B,D; Appendix A) and in the cortex (Figure 4C,D) of NPC1 vs. wt mice, we observed similar significant decreases of sorLA immunostaining in neuronal soma as in NPC1 cerebellum at 4-weeks of age. In addition, in the hippocampus, a shift in Vps35 staining was observed between the mossy fibers in the wt mice to pyramidal neurons in the CA3 region of NPC1 mice (Figure 4B). Further analysis of the hippocampal CA1 and dentate gyrus regions revealed a similar and significant decrease of sorLA staining and accumulation of Vps35 in NPC1 vs. wt mice (Appendix A). The observed changes of sorLA and Vps35 immunostaining in NPC1 vs. wt hippocampus’ and cortices were already detected at 4-weeks of age (Figure 4B–D, respectively), but became more apparent with the progression of the disease in NPC1 mice at 7 and 10-weeks of age. In particular, it could be observed that those cortical neurons that had the lowest signal for sorLA in NPC1 brains had the strongest Vps35 accumulation and vice versa (Figure 4C). The observed altered immunostaining of sorLA and Vps35 in NPC1 vs. wt brains already at presymptomatic stage (Figure 4) indicates that retromer dysfunction occurs early in the course of NPC disease and may contribute to downstream pathological processes. Furthermore, the levels of Vps35 and sorLA did not differ between NPC1 and wt-mouse brains in all three brain regions analyzed at both 4 and 10-weeks of age (results not shown). Overall, our results suggest that retromer impairment in NPC is an early event that is characterized by disturbed retromer trafficking and distribution.

Using discontinuous sucrose gradient and endosomal fractionation, we further analyzed retromer distribution within the endolysosomal pathway between 10-week old NPC1 and wt mouse brains. We reasoned that at 10-weeks of age, at the end-stage of the disease, we will identify the most profound changes of retromer in NPC1 mouse brains. All endolysosomal markers analyzed (EEA1, Vps35, sorLA, Rab7 and LAMP1) showed broader distribution throughout the gradient in NPC1 vs. wt-mouse cerebella (Figure 5A), suggesting a mixed molecular identity of endocytic organelles. In particular, we observed a shift in the distribution of Vps35 and sorLA in fraction 2 of NPC1 mouse cerebella compared to wt cerebella, which represents late endosomal/lysosomal fraction (Figure 5A). This shift may imply that due to NPC1-loss and disturbed cholesterol homeostasis retrograde, trafficking via retromers is mistargeted into lysosomes and that this may additionally contribute to the disturbed endolysosomal function. Moreover, the analysis of the BACE1-exclusive substrate Sez6L [52,53] revealed its prominent shift into early endocytic fraction 5 in NPC1 in contrast to wt cerebella (Figure 5B). Since fraction 5 in NPC1 brains was strongly positive for EEA1, Vps35 and sorLA markers, we propose that previously described enhanced BACE1-mediated proteolysis of Sez6L [48], as well as of other substrates, in NPC1 vs. wt mouse brains likely involves its accumulation within early endosomes—a prime site of the BACE1 cleavage (Figure 5B). Quantification of the levels of unesterified free cholesterol confirmed their significant increase in late endosome fractions in NPC1 vs. wt cerebella (Figure 5C), further supporting that the isolated fractions were indeed late endosomes/lysosomes.

### 2.3. NPC1-Mouse Primary Neurons Show Retromer Transport Defect

Since we observed changes in the retromer in the NPC1 mouse brains already at the presymptomatic stage of the disease, at 4-weeks of age, we analyzed retromer localization in the NPC1 vs. wt-mouse primary neuronal cultures in more detail. To analyze very early defects of retromer distribution, we isolated neurons from the brain region, such as the hippocampus, that is presumed to be protected or less/later affected by the NPC pathology. Indeed, although hippocampal neurons in NPC disease show less vulnerability to neurodegeneration in comparison to Purkinje neurons, our results of immunohistochemistry of NPC1 mouse brains demonstrated that hippocampal neurons, together with cortical neurons, also have altered retromer distribution, indicating their pathological alterations in NPC. Thus, we isolated hippocampal and cortical neurons from NPC1 and wt P0 pups and monitored subcellular localization and levels of Vps35, Vps26 and/or sorLA, respectively.

We observed that NPC1-hippocampal neurons show free cholesterol accumulation in their soma and enlarged lysosomes (Figure 6A,D)—the two main features of NPC disease. Indeed, LAMP1 staining revealed significantly increased enlarged lysosomes in the NPC1 vs. wt hippocampal neurons (Figure 6A,D). Furthermore, retromer protein Vps35 showed a significantly increased size of puncta in NPC1 vs. wt-hippocampal neurons (Figure 6A–D). In addition, the size of the puncta of TfR and LAMP1-positive vesicles appeared to increase in NPC1 vs. wt hippocampal neurons, both in the soma and processes (Figure 6A–D). Furthermore, we observed significantly enhanced colocalization of Vps35 and all the analyzed endocytic markers in NPC1 compared to wt neurons (Figure 6A–C,E). Filipin staining was found to be significantly increased in the soma with a slight but no significant decrease in the axons (Figure 6A,D), which is in line with the previously suggested redistribution of accumulating free cholesterol between soma and the processes [35]. The observed altered trafficking of Vps35 in the soma and/or processes of NPC1 hippocampal neurons did not involve the altered expression of Vps35 protein, as well as of retromer proteins Vps26 and sorLA (Appendix A). Thus, we conclude that altered retromer trafficking in NPC is most likely due to endolysosomal transport defects. The compromised transport within NPC1 neuronal processes was further confirmed by assessing axonal swellings. Primary mouse NPC1 hippocampal neurons, stained with a neuronal-specific marker TUJ-1, showed axonal swellings in which LAMP1 and Vps35-positive endolysosomal vesicles were also accumulating (Figure 7). These results suggest that a traffic jam of retromers and of endolysosomal vesicles within NPC1-neuronal processes may cause axonal swellings and are indicative of early pathological changes leading to subsequent neuronal malfunction.

### 2.4. Retromer Impairment Is Dependent on Cholesterol Levels

To identify the offending lipid triggering the retromer impairment, we modulated cholesterol levels and assessed its effect on retromer distribution. We reasoned that cholesterol-lowering strategies that rescue a prime pathological hallmark of NPC disease—accumulation of free cholesterol due to NPC1/2 dysfunction—may revert the retromer trafficking defect in NPC models and that cholesterol-loading in wt cells may, in contrast, cause retromer mistrafficking similar to that in NPC models.

First, we depleted cholesterol levels in *NPC1*-null cells using four different approaches: LPDS-, LPDS+lovastatin, methyl-β cyclodextrin (MβC)-acute and MβC-chronic treatment. These cholesterol-lowering strategies were used in vitro to decrease uptake of exogenous lipids (LPDS-treatment), to inhibit enhanced cholesterol biosynthesis (statin-treatment, which is the most widely used medication to decrease cholesterol levels) and to extract membrane-embedded cholesterol (MβC-treatment) [46]. All four treatments of cholesterol depletion in *NPC1*-null cells significantly decreased both the total and free cholesterol levels to levels similar or lower than in CHOwt (Appendix A). Filipin staining confirmed significantly decreased intracellular accumulation of unesterified cholesterol in treated *NPC1*-null cells (Appendix A). In addition, a Lysotracker probe revealed an increased number and the size of lysosomal puncta in *NPC1*-null cells that were significantly decreased upon cholesterol depletion (Appendix A). These results confirmed that cholesterol-depleted approaches successfully rescued free cholesterol accumulation and lysosomal dysfunction in *NPC1*-null cells. While the protein levels of Vps26, Vps35 and endolysosomal markers were not altered upon cholesterol depletion in *NPC1*-null cells (Appendix A), immunostaining revealed that in treated *NPC1*-null cells, enlarged retromer vesicles were lost and that Vps26 and Vps35 colocalized in smaller puncta, similar to that observed in CHOwt (Figure 8A,D,E). Further analysis of Vps35 colocalization with organelle markers in cholesterol depleted vs. untreated *NPC1*-null cells demonstrated that increased size of TfR and LAMP1-positive puncta and their increased colocalization with Vps35 was lost upon cholesterol depletion that now resembled that as in CHOwt cells (Figure 8B–E). To summarize, these results implicate the rescue of retromer and endolysosomal transport system upon cholesterol depletion in *NPC1*-null cells (Figure 8A–E).

Secondly, to load cholesterol in vitro in CHOwt cells, we used two different approaches: U18666A-treatment, which mimics intracellular cholesterol accumulation within late endosomes/lysosomes as in *NPC1*-null cells, and MβC-cholesterol complex treatment, which causes increased cholesterol in all cellular membranes (plasma membrane and intracellular membranes) [46]. These cholesterol-loading approaches in CHOwt cells caused significantly increased total and free cholesterol levels (Appendix A), accumulation of unesterified cholesterol in late endosomes and lysosomes (U-treatment) and enhanced cholesterol staining within the cell (MβC-treatment) (Appendix A). In addition, Lysotracker staining revealed the increased size of the lysosomal puncta upon cholesterol loading of CHOwt cells (Appendix A), indicating lysosomal impairment. Increased cholesterol levels did not alter the levels of retromer and endolysosomal proteins in CHOwt cells (Appendix A). However, it caused significantly increased size of Vps35 and Vps26-positive vesicles, similar to that as in *NPC1*-null cells (Figure 9A–F), as well as increased size of the puncta of endocytic vesicles, EEA1 for early (Figure 9B,F), TfR for recycling (Figure 9C,F), Rab7 for late endosomes (Figure 9D,F) and LAMP1 for lysosomes (Figure 9E,F). Moreover, we observed significantly increased colocalization of Vps35 with EEA1 (Figure 9B,G), TfR (Figure 9C,G), Rab7 (Figure 9D,G) and LAMP1-vesicles (Figure 9E,G). Overall, these in vitro cholesterol depletion and loading approaches show that retromer trafficking is modulated by cholesterol levels, indicating that there are molecular events associated with cholesterol dyshomeostasis that regulate subcellular retromer distribution.

Lastly, we tested whether cholesterol-lowering treatments in vivo would rescue retromer distribution in NPC1-mouse brains to that as in wt mice. We pharmacologically treated 10-week old NPC1-deficient mice with MβC for five days or statin fluvastatin for seven days. Indeed, it has been previously shown that cholesterol depletion using cyclodextrins (HPβCD) rescues the lethality of NPC1 mice and improves pathological hallmarks of NPC disease [39]. We also treated NPC1 mice with the most common cholesterol-lowering medication, statin (fluvastatin), which inhibits cholesterol biosynthesis. The cortices of untreated and treated NPC1 mice were analyzed by immunohistochemistry. Strikingly, reduced sorLA signal seen in NPC1-deficient mice was partially rescued by cholesterol depletion (Figure 10). Moreover, the cells that now exhibit sorLA immunoreactivity in MβC/fluvastatin-treated NPC1-deficient mice also show reduced Vps35 accumulation (Figure 10A). Quantification of the obtained immunohistochemistry signals confirmed the upregulation of the sorLA staining and, in parallel, downregulation of the Vps35 signal upon cholesterol depletion in vivo in NPC1 mice (Figure 10B). These results indicate that Vps35 and sorLA distribution is dependent on cholesterol levels and that, even at the end-stage of NPC disease at 10-weeks of age, impaired retromer localization in NPC1-mice can be partially rescued by cholesterol depletion. Thus, we show that in vivo cholesterol-lowering approaches are beneficial against retromer impairment, further supporting the mechanistic link between cholesterol dyshomeostasis and altered retromer transport.

## 3. Discussion

Retromer dysfunction is linked to several neurodegenerative disorders, including the most common AD and PD [11]. Indeed, new findings suggest that retromer defects could be a central signaling hub for several pathological features, such as protein aggregation (of Aβ, α-synuclein and tau), loss of synapses, neuronal dysfunction and neuroinflammation (microglia impaired function) [54]. Since all these features are characteristic of a rare lipid storage and neurodegenerative disorder Niemann-Pick type C (NPC), in this work, we analyzed if retromer trafficking (dys)function is involved in the pathogenesis of NPC disease. To address this, we used several NPC models: CHO *NPC1*-null cell line, NPC1-deficient mouse primary neurons and NPC1-deficient mice. In line with the recently demonstrated link between endolysosomal cholesterol homeostasis and retromer function [55], here we show, for the first time, that retromer transport is impaired in NPC disease. Moreover, a retromer defect was observed early in the presymptomatic stage and was rescued by cholesterol depletion, indicating that retromer alteration in NPC is an early event in the disease pathogenesis and is dependent on intracellular cholesterol accumulation.

We identified that retromer impairment in NPC models is characterized by altered distribution of the retromer core components Vps26 and Vps35 and of the retromer receptor sorLA. Indeed, the observed retromer defect in *NPC1*-null cells and in NPC1-mouse primary neurons was comparable to that recently described in Arf6 knockout (KO) mouse embryonic fibroblasts (MEFs), that also showed no alterations in retromer complex levels but the accumulation of retromers in alveolar-shaped enlarged vesicles, suggesting disturbed retromer transport. Similar to findings in Arf6 KO MEFs, the accumulating enlarged retromers mainly colocalized with recycling endosomal marker and partially with early endosomes in *NPC1*-null cells and in NPC1-mouse primary neurons. Interestingly, retromer deficiency via Vps35 knockdown (KD) in HeLa cells caused an increase in cholesterol levels and an accumulation of cholesterol in late endosomes/lysosomes [55], similar to that in Arf6 KO and *NPC1*-null cells. These findings indicate that there is a bidirectional link between retromer function and intracellular cholesterol transport and that NPC retromers display an Arf6-like phenotype, suggesting a shared mechanism of retromer defect. This is further supported by the fact that over-expression of Arf6 reduced cholesterol accumulation in NPC1-deficient cells [56]. Furthermore, a small GTPase Arf6 was identified to control retromer traffic and intracellular cholesterol distribution via a phosphoinositide-based mechanism that involves the accumulation of phosphatidylinositol-4-phosphate (PI4P) in endosomes and that reducing PI4P levels in Arf6 KO MEFs, through independent mechanisms, rescues aberrant retromer tubulation and cholesterol mistrafficking [55]. The PI4P levels were recently found to also be increased in NPC1 KO and NPC1 mutant cells [57], due to enhanced recruitment of PI4-kinases (PI4KIIα and PI4KIIIβ) at Golgi and lysosomal membranes, and that this can be rescued by the inhibition or knockdown of either key phosphoinositide enzymes or their recruiting partners. While molecular details of PI4P increase and PI4KIIα and PI4KIIIβ redistribution in NPC are still to be elucidated, Kutchukian and colleagues [57] have demonstrated that in NPC1 KO and NPC1 mutant cells enhanced palmitoyl acyltransferase DHHC3-dependent palmitoylation and increased levels of multi-functional adaptor protein acyl-CoA-binding domain-containing 3 (ACBD3) are likely involved in this process. Thus, the recent results in Arf6 KO MEFs and in *NPC1*-null/mutant cells demonstrate that the altered distribution of PI4-kinases (PI4KIIα and PI4KIIIβ) and/or accumulation of PI4P could be central to altered intracellular cholesterol trafficking and retromer dysfunction in NPC disease. These findings are fully in line with retromer impairment in primary mouse NPC1 neurons and NPC1 mouse brains described in this work, indicating that the PI4P pathway could be relevant to NPC pathology and that mechanistic studies elucidating intracellular trafficking defects in NPC deserve more attention. Since we showed that retromer impairment in NPC disease models is dependent on increased cholesterol levels, we speculate that PI4P levels and/or redistribution of PI4-kinases (PI4KIIα and PI4KIIIβ) could be linked to cholesterol dyshomeostasis, which needs to be further investigated. Furthermore, approaches that lead to reduced PI4P levels and/or decreased activity of PI4-kinases (PI4KIIα and PI4KIIIβ) may represent a novel strategy to rescue impaired retromer transport and altered intracellular cholesterol trafficking in NPC disease and, potentially, in other lipid storage, as well as more common neurodegenerative disorders. Since the above-mentioned mechanistic findings were obtained in distinct cell lines (MEFs, HEKs, HeLa) and not in neuronal and glial cells or NPC mouse/patients brain tissue, further studies are necessary to confirm whether similar defects in the subcellular redistribution of PI4-kinases (PI4KIIα and PI4KIIIβ) and in PI4P homeostasis occurs in NPC disease-affected brain cells/tissues. Although in this work, we analyzed retromer defects in primary mouse NPC1 neurons and in neurons within distinct NPC1 mouse brain regions, it also would be interesting to investigate retromers in NPC myeloid cells where we recently reported defects in endolysosomal trafficking that were triggered by increased cholesterol [36]. In particular, proteomic analysis of NPC microglia revealed increased levels of PI4KIIα that were among the earliest defects detected in presymptomatic NPC mice (already at P7, even more pronounced at 8 weeks). Rab7a was also found to increase in our proteomic analysis of NPC microglia. Furthermore, TREM2, a microglial protein that is trafficked via retromer and is involved in the development of AD [24,58], was also increased in NPC microglia. Thus, we reason that astrocytes and microglia, which are activated in NPC brains, are likely to exhibit retromer defects as well.

Using NPC1 mouse brains (cerebellum, cortex and hippocampus) we further confirmed altered retromer trafficking already at the presymptomatic stage of NPC disease, at 4-weeks of age. It was characterized by decreased expression of sorLA in neuronal soma, followed by the accumulation of Vps35 as a potential compensatory mechanism. Our finding of altered distribution of retromer receptor sorLA in NPC mouse brains, which is involved in retrograde trafficking of the key AD proteins APP and BACE1, and is genetically linked to both rare familial and a more common complex form of AD, provides further support for the link between AD and NPC. Since sorLA is involved in retrograde trafficking of APP and BACE1 out from early endosomes, thus decreasing amyloidogenic APP pathway and Aβ formation, we reasoned that its impairment in NPC may be responsible for enhanced amyloidogenic processing of APP by BACE1 [47,48], as well as for increased proteolysis of other BACE1 substrates that we have previously described in NPC1 mouse brains [48]. Interestingly, both AD and NPC show similar retromer redistribution in mouse primary neurons with decreased retromer levels in neuronal soma and retromer accumulation in neuronal processes. Thus, we conclude that altered retromer trafficking is central to retromer dysfunction in both AD and NPC disease.

Our results of manipulating cholesterol levels in vitro in *NPC1*-null cells and in vivo in NPC1 mice further support the recently established link between intracellular cholesterol distribution and retromer function. Indeed, we showed that in *NPC1*-null background (in *NPC1*-null cells and in NPC1-deficient mouse brains), retromer defects can be rescued by cholesterol depletion and vice versa that cholesterol overload in wt cells may induce retromer impairment. This finding indicates that the molecular mechanism that leads to the altered function of a retromer complex in NPC disease is dependent on intracellular cholesterol accumulation. Based on the findings of Marquer and colleagues [55] and Kutchukian and colleagues [57], we speculate that altered phospholipid signaling may also contribute to cholesterol-dependent retromer impairment in NPC disease. Future studies are needed to confirm this and to analyze whether neuronal/non-neuronal vulnerability (of astrocytes and/or microglia) is associated with alterations of any specific phospholipid species in NPC.

The observed defect in retromer distribution at the presymptomatic stage in NPC mouse brains further indicates that a retromer alteration is an early event in the course of NPC disease, which may drive and/or contribute to the pathogenesis of NPC. Retromer accumulation in early and most profoundly in recycling endosomes in *NPC1*-null cells and along axons in axonal swellings in NPC1 mouse primary neurons indicates that altered retromer distribution and its altered axonal transport may contribute to synaptic loss and neuronal degeneration in NPC disease. Although further studies are necessary to elucidate the role of retromer dysfunction in the pathogenesis of NPC disease and the molecular details that trigger retromer impairment, our findings suggest that retromer dysfunction could be a common mechanism in the pathogenesis of both complex and rare neurodegenerative disorders and that retromer-targeted therapy may be beneficial in multiple etiologically distinct neurodegenerative disorders, including NPC. Thus, in the future, it would be important to test whether small compounds/pharmacological chaperons that were shown to rescue retromer defect in multiple neurodegenerative disorders [29,31], as well as approaches that lead to reduced PI4P levels and/or to decreased activity of PI4-kinases (PI4KIIα and PI4KIIIβ) [55,57], may play a role in the amelioration and/or reversal of NPC disease.

## 4. Material and Methods

### 4.1. Cell Culture and Primary Neurons

Chinese hamster ovary wild type (CHOwt) cells and CHO-*NPC1*-null cells (kindly provided by Dr. Daniel Ory) were maintained in DMEM:F12 medium (1:1) containing 0.5 mM Na-pyruvate supplemented with 10% FBS, 2 mM L-glutamine and antibiotic/antimycotic (all from Sigma-Aldrich, St. Louis, MO, USA).

Primary neuronal cultures of hippocampal neurons were prepared from *NPC1^+/+^* (wt) and *NPC1^−/−^* (NPC1) mouse pups at postnatal day 0 (P0). Hippocampal neurons were isolated, plated on poly-L-lysine (0.5 mg/mL, Sigma-Aldrich, St. Louis, MO, USA) coated glass coverslips (Marienfeld, Lauda-Königshofen, Germany) or tissue culture dishes (Nunclon, Thermo Fischer Scientific, Waltham, MA, USA) and grown in Neurobasal medium complemented with 2% B27 and 2 mM L-glutamine (all from Gibco, Thermo Fisher Scientific, Waltham, MA, USA) as previously described [59].

### 4.2. NPC1 Mouse Model

A mouse model of NPC disease, the BALB/cNctr-*Npc1m1N*/J (stock number 003092), was obtained from the Jackson Laboratory, Bar Harbor, Maine, USA. For immunohistochemistry, they were housed at the JSW Lifesciences GmbH/QPS Austria GmbH animal facility in accordance with the EU Directive 2010/63/EU for animal experiments. For primary neuronal cultures, mice were housed at the facility for laboratory animals (https://www.irb.hr/eng/Divisions/Division-of-Molecular-Medicine/Facility-for-laboratory-animals (accessed on 1 November 2021)) at the Ruder Boskovic Institute in accordance with the relevant national guidelines: Animal Protection Act (Law on the Welfare of Animals, National Gazette #102, 2017 and the regulations on the protection of animals used in experiments or other scientific purposes, National Gazette #39, 2017). This research was approved by the Ethic Committee of the Ruder Boskovic Institute and by the Ministry of Agriculture of the Republic of Croatia (UP/I-322-01/15-01/42 and UP/I-322-01/17-01/15).

For this experiment, mice of wt and NPC1 genotypes at different disease stages (age 4, 7, and 10 weeks) [60] were used (6 animals per group, in total 54 animals). For isolation of primary mouse neurons at postnatal day 0 (P0), 60 mouse pups were used.

The animals were housed at 22 ± 1 °C and 50–70% humidity with a 12/12 h light/dark cycle photoperiod. They were maintained on a formulated commercial pellet diet and water provided ad libitum. Since *NPC1^−/−^* mice are not fertile, female and male *NPC1^+/−^* mice were mated to generate *NPC1^−^*^/*−*^ (NPC1) and *NPC1^+/+^* (wt) mice. Mice were genotyped according to already published protocols (http://jaxmice.jax.org/strain/003092.html (accessed on 1 November 2021)).

All mice were sacrificed by cervical dislocation and perfused with 0.9% saline solution. Brains were removed and hemisected, and right hemispheres were immersion fixed in fresh 4% paraformaldehyde in phosphate-buffered saline (PBS) for 24 h and then transferred first to a 15% sucrose solution and then to a 30% sucrose solution for 24 h to ensure cryoprotection. The next day hemispheres were frozen in liquid isopentane and embedding matrix (Shandon M1-Embedding Matrix, Thermo Fischer Scientific, Waltham, MA, USA) and stored at −80 °C until cutting.

### 4.3. Antibodies

For retromer detection, antibodies against Vps26 (Abcam, Cambridge, UK), Vps35 (Novus Biologicals, Littleton, CO, USA) and sorLA (from Merck Millipore, St. Louis, MO, USA, for immunocytochemistry and BD, Franklin Lakes, NJ, USA, for Western blot analysis) were used. The following antibodies were used as organelle markers: TGN46 (Abcam, Cambridge, UK), EEA1 (Cell Signaling Technology, Danvers, MA, USA), Rab7 (Abcam, Cambridge, UK), transferrin receptor (Life Technologies, Waltham, MA, USA) and LAMP1 (Sigma, St. Louis, MO, USA and Santa Cruz, Dallas, TX, USA). For APP labelling N-terminal antibody (APP-NT, 22C11, Merck Millipore, St. Louis, MO, USA) and C-terminal antibody (APP-CT, Y188, Abcam, Cambridge, UK) were used. Antibodies against β-actin were purchased from Abcam (Cambridge, UK), TUJ-1 from Covance (Denver, PA, USA) and calbindin from Swant (Burgdorf, Switzerland).

### 4.4. Cell Lysate Preparation

For Western blot analysis, cell lysates were prepared as follows: cells were washed three times in PBS and homogenized in RIPA buffer (50 mM Tris pH 8.0, 150 mM NaCl, 5 mM EDTA, 1% NP40, 0.1% SDS, 0.5% sodium deoxycholate) containing protein inhibitor cocktail (Roche Applied Science, Basel, Switzerland) and centrifuged at 16,000× *g* for 10 min at 4 °C. Total protein concentration was measured using a commercially available Pierce BCA Protein Assay Kit (Thermo Fischer Scientific, Waltham, MA, USA) according to the manufacturer’s protocol.

### 4.5. Western Blot

Cell lysates were mixed with 6 × sample buffer (60% glycerol, 12% SDS, 3% DTT, 1/8 *v*/*v* 0.5 M Tris pH 6.8, bromophenol blue) and heated at 70 °C for 10 min. Equivalent amounts of protein were loaded onto Tris-Glycine gels. After SDS-PAGE, proteins were transferred onto PVDF membrane (Roche Applied Science, Basel, Switzerland) and incubated in primary antibody solution, followed by washing and HRP-conjugated secondary antibody (Bio-Rad, Hercules, California, USA) incubation. Proteins were visualized by chemiluminescence using POD chemiluminescence blotting substrate (Roche Applied Science, Basel, Switzerland) on a documentation system from UVItec, Cambridge, UK. To control the same amount of protein, a loaded antibody against β-actin (Sigma-Aldrich, St. Louis, MO, USA) was used. Protein signals were quantified using the ImageJ software (National Institutes of Health, Bethesda, MD, USA). 

To analyze the isolated endosome fractions by Western blotting, the samples of the wt and NPC1 mouse brains were normalized prior to ultracentrifugation and endosome fractionation by loading the same amount of the postnuclear supernatant (PNS) into the gradient. Upon ultracentrifugation, equal volumes of isolated fractions were loaded and analyzed by Western blotting. Protein signals were quantified in each fraction using the ImageJ software (National Institutes of Health, Bethesda, MD, USA). The obtained signal intensity represents the amount of the analyzed vesicular marker in each fraction.

### 4.6. Immunocytochemistry and Confocal Microscopy

For immunocytochemistry, CHO cells were grown on coverslips and fixed in 4% paraformaldehyde/sucrose, permeabilized with 0.2% saponin and blocked in 4% donkey serum (all from Sigma-Aldrich, St. Louis, MO, USA). Immunostaining was performed with primary antibodies at 4 °C overnight, following 1 h of incubation with secondary anti-rabbit-Alexa488 or anti-mouse-Alexa594 or anti-goat-Alexa647 antibodies from donkeys (Molecular Probes, Invitrogen, Waltham, MA, USA) and stained with filipin (Sigma-Aldrich, St. Louis, MO, USA), as described by [61]. The cells were mounted (Fluoromount, Sigma-Aldrich, St. Louis, MO, USA) and viewed by laser scanning confocal microscope Leica SP8 X FLIM (Leica, Wetzlar, Germany).

Immunocytochemistry on mouse primary hippocampal neurons was performed as follows: neurons were fixed with 4% paraformaldehyde/sucrose, quenched in 50 mM ammonium chloride and permeabilized with 0.1% Triton X-100 for 3 min, and proceeded with the staining as described above.

For lysosome function analysis, live cells grown on coverslips were stained with LysoTracker Red DND-99 (Molecular Probes, Thermo Fischer Scientific, Waltham, MA, USA) at a final concentration of 200 nM for 80 min at 37 °C followed by fixation, Hoechst staining (Sigma-Aldrich, St. Louis, MO, USA) and confocal microscope analysis, excitation/emission: 577/590 nm.

Quantification of the obtained immunofluorescent signals was performed using ImageJ software (National Institutes of Health, Bethesda, MD, USA).

### 4.7. Immunohistochemistry of Mouse Brain Cryosections

The paraformaldehyde-fixed and sucrose-cryoprotected frozen right hemispheres from wt and NPC1 mice were used to prepare 10 μm thick sagittal cryosections (Leica CM 3050S cryotome, Wetzlar, Germany). For immunohistochemistry, cryosections were dried in the hood for 1 h, permeabilized in TBS-T (50 mM TBS, pH 8.0, 0.5% Triton X-100, Sigma-Aldrich, St. Louis, MO, USA) for 30 min and blocked in 4% donkey serum (Sigma-Aldrich, St. Louis, MO, USA) in TBS-T for 1 h. The sections were subsequently incubated with primary antibodies diluted in 4% donkey serum in TBS-T overnight. After a 3 h incubation with secondary antibody anti-rabbit-Alexa488, anti-mouse-Alexa594 or anti-goat-Alexa647 (from donkey, Molecular Probes, Invitrogen, Waltham, MA, USA) and Hoechst to counterstain nuclei (Sigma-Aldrich, St. Louis, MO, USA), sections were mounted (Fluoromount, Sigma-Aldrich, St. Louis, MO, USA). Confocal images were acquired on an inverted laser scanning confocal microscope Leica TCS SCP8 with the software LAS X (Leica, Wetzlar, Germany), and additional image processing and quantification were performed using ImageJ software (National Institutes of Health, Bethesda, MD, USA).

### 4.8. Cholesterol Depletion/Loading

To deplete cholesterol, *NPC1*-null cells were incubated for 48 h in the medium containing 10% lipoprotein deficient serum (LPDS, Cocalico Biologicals, Reamstown, Pennsylvania, USA) instead of FBS (LPDS treatment) [46] or in 10% LPDS medium containing 4 μM lovastatin and 0.25 mM mevalonate (Sigma-Aldrich, St. Louis, MO, USA) for statin treatment [62]. In addition, methyl-β-cyclodextrin (MβC, Sigma-Aldrich, St. Louis, MO, USA), to acutely or chronically deplete cholesterol, was used. For acute depletion, cells were treated with 10 mM MβC in 10% LPDS media for 1 h, after which was kept in 10% LPDS media for 24 h. For chronic depletion, cells were treated with 1 mM MβC in 10% LPDS media for 24 h, as described previously [63].

To load cholesterol, CHOwt cells were treated with 100 μM MβC-cholesterol complex (Sigma-Aldrich, St. Louis, MO, USA) for 4 h after 24 h incubation in 10% LPDS medium [64]. To mimic NPC phenotype, CHOwt cells were treated with 3 μg/mL U18666A (Sigma-Aldrich, St. Louis, MO, USA) for 48 h [65]. The levels of total and free cholesterol in cell lysates were determined using the Amplex Red Cholesterol Assay Kit (Molecular Probes, Invitrogen, Waltham, MA, USA) and measured on a fluorometer (Fluoroskan Ascent FL, Thermo Fisher Scientific, Waltham, MA, USA).

To deplete cholesterol in vivo, NPC1 mice were treated with MβC or fluvastatin (*n* = 3) or with placebo (5% DMSO in 0.9% saline solution). All the animals were sacrificed 5 hours after the last treatment. MβC (S0940, 4000 mg/kg/day, Sigma Aldrich, St. Louis, MO, USA) was injected subcutaneously, starting on 24th, 45th or 66th day of age, and the treatment lasted for 5 days. Fluvastatin (#3309, 5 mg/kg/day, Tocris Bioscience, Bristol, UK) was applied by cannula injection once a day, starting on the 22nd, 43rd or 64th day and treatment lasted for 7 days.

### 4.9. Endosome Fractionation

Endosome fractionation was performed using a step gradient centrifugation by a previously described protocol [66] with some modifications. Briefly, cerebella of 10-week old wt and NPC1 mice were homogenized in homogenization buffer (250 mM sucrose, 3 mM imidazole, 1 mM EDTA, pH 7.4) containing protease inhibitor cocktail and phosphatase inhibitor (Roche Applied Science, Basel, Switzerland) in a Dounce homogenizer. The homogenate was passed 10 times through a 23-gauge needle and centrifuged first at 1000× *g* for 10 min and then at 3000× *g* for 10 min. The equal amount of PNS of wt and NPC1-mouse brains were loaded at the bottom of an ultracentrifuge tube (14 × 89 mm, Beckman Coulter, Brea, California, USA) and adjusted to 40.6% sucrose in 3 mM imidazole (Fluka, Sigma-Aldrich, St. Louis, MO, USA), pH 7.4 and 1 mM EDTA (Kemika, Zagreb, Croatia) by mixing with 62% sucrose in 3 mM imidazole (pH 7.4) and 1 mM EDTA. Three milliliters of PNS mixture were overlaid with 4.5 mL of 35%, 3 mL of 25% and 1 mL of 8% sucrose in 3 mM imidazole (pH 7.4) and 1 mM EDTA. After centrifugation (3 h, 4 °C, 210,000× *g*, rotor SW41.Ti), eleven 1 mL fractions were collected from the top of the gradient. Fractions 1 and 2 (8% and 25% interface) represent late endosome fraction (LE), fractions 4 and 5 (25% and 35% interface) early endosome fraction (EE) and fractions 8 and 9 (35% and 40.6% interface) heavy fraction (HF). Each fraction was diluted 1:1 in a solution containing 3 mM imidazole (pH 7.4) and 1 mM EDTA and centrifuged at 100,000× *g* for 1 h, rotor 50.4 Ti. After centrifugation, organelle pellets were diluted in an equal volume of RIPA buffer and aliquots were analyzed by Western blotting. The levels of free unesterified cholesterol in the late endosome and early endosome fractions were determined using the Amplex Red Cholesterol Assay Kit (Molecular Probes, Invitrogen, Waltham, MA, USA).

### 4.10. Statistical Analysis

Statistical analysis of the data was performed using SPSS statistical package for Windows, v17. Before all analyses, samples were tested for normality of distribution using the Shapiro–Wilk test. If the data were not following a normal distribution, an appropriate transformation of the data was made, and further analyses were performed on transformed data. If the data did not follow normal distribution after transformation, non-parametric Kruskal–Wallis H-test, with a pairwise comparison between groups using Dunn’s procedure with Bonferroni correction for multiple comparisons was applied.

For the determination of statistically significant differences between the means of three or more groups, one-way ANOVA with appropriate post-hoc analyses were used. To determine if a difference exists between the means of two independent groups on a dependent variable, an independent-samples *t*-test was performed. The statistical significance of the tests was set at *p* < 0.05.

## Figures and Tables

**Figure 1 ijms-22-13256-f001:**
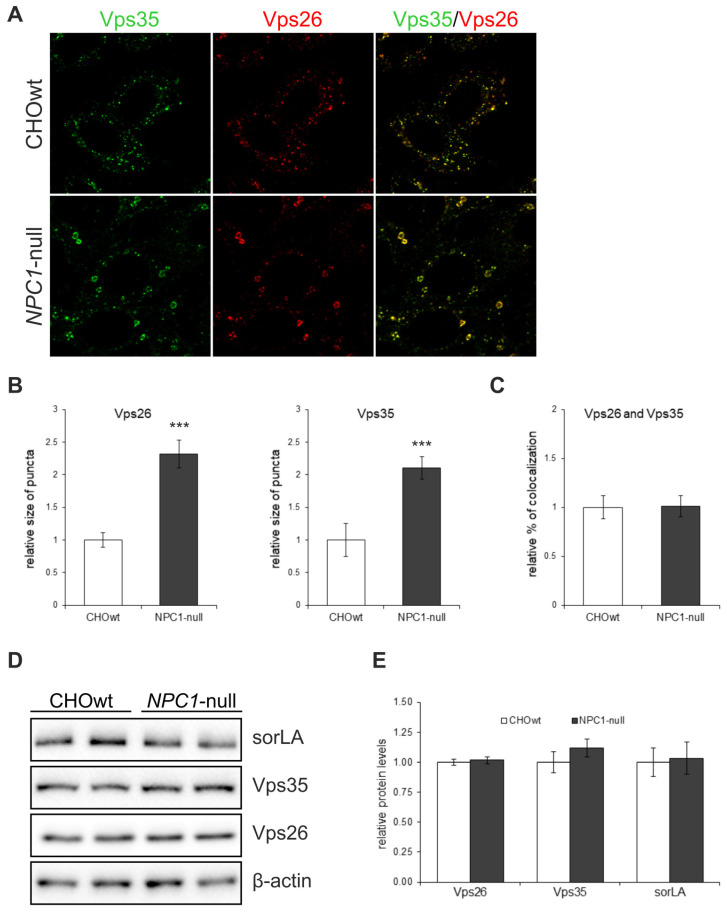
Retromer proteins accumulate in enlarged vesicles at the periphery in *NPC1*-null cells compared to CHOwt. (**A**) The cells were grown in 10% FBS in DMEM/F12 media, fixed and stained for Vps26 (green) and Vps35 (red). Cells were analyzed by confocal microscopy, and fluorescent signals were quantified using ImageJ software. (**B**) Size of Vps26 or Vps35-positive puncta and (**C**) colocalization of Vps26 and Vps35 were compared between CHOwt and *NPC1*-null cells, *** *p* < 0.001, *n* = 30 cells per group from three different experiments. (**D**,**E**) The levels of retromer proteins Vps26 and Vps35 and retromer receptor sorLA, analyzed by western blot, did not reveal any differences in the protein levels. Protein signals were quantified using the ImageJ software. The statistical significance of the tests was set at *p* < 0.05. The data are shown as mean ± SD normalized to control and represent data from three different experiments in duplicates, *n* = 6 per group. Scale bar—20 µm.

**Figure 2 ijms-22-13256-f002:**
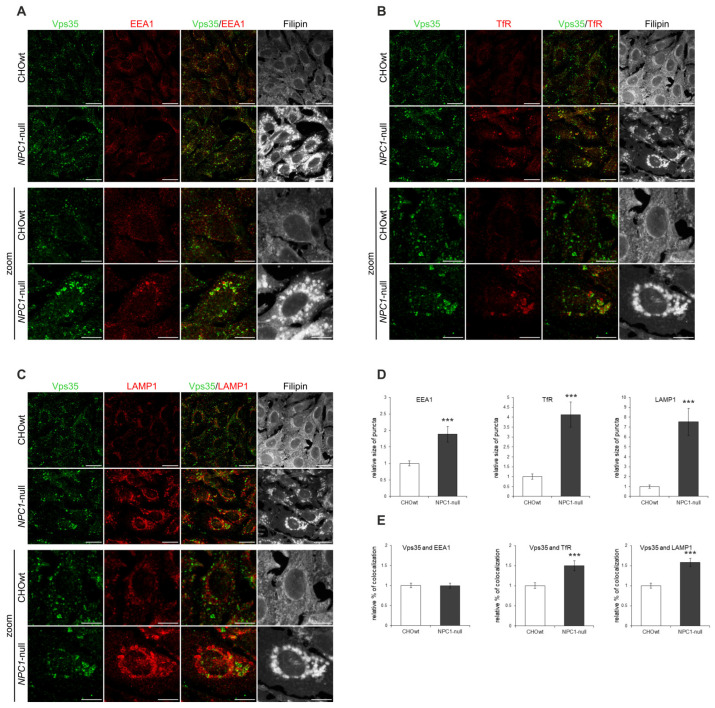
Retromer trafficking within the endolysosomal pathway is altered in *NPC1*-null cells. Trafficking of the retromer protein Vps35 in CHOwt and *NPC1*-null cells was monitored by immunofluorescence (**A**–**C**), and the obtained signals were quantified using ImageJ software (**D**,**E**). The cells were grown in 10% FBS in DMEM/F12 media, fixed and stained for Vps35 (green) and (**A**) early endosome marker EEA1 (red), (**B**) recycling endosome marker transferrin receptor (TfR, red) and (**C**) lysosome marker LAMP1. Filipin was used to stain free cholesterol. (**D**) Size of EEA1, TfR and LAMP1-positive puncta and (**E**) colocalization of Vps35 with EEA1, TfR and LAMP1 were compared between CHOwt and *NPC1*-null cells, *** *p* < 0.001. The data are shown as mean ± SD normalized to control, *n* = 30 cells per group from three independent experiments. Scale bar—20 µm (unzoomed images) and 10 µm (zoomed images).

**Figure 3 ijms-22-13256-f003:**
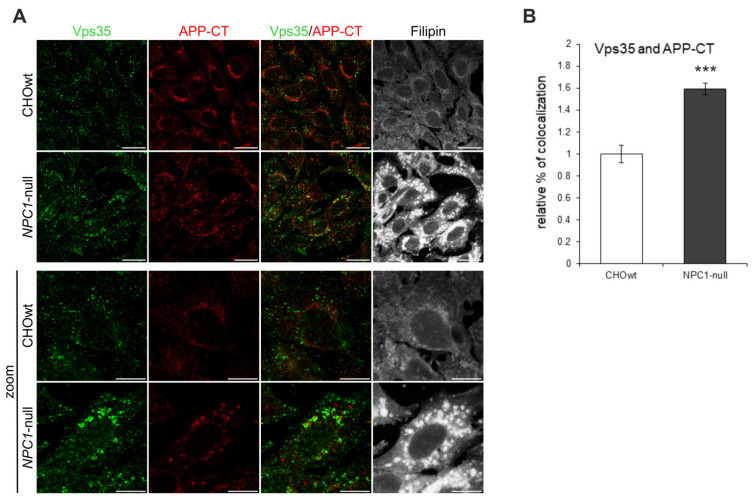
Colocalization of Vps35-positive vesicles and APP-CT is increased in *NPC1*-null cells. (**A**) The cells were grown in 10% FBS in DMEM/F12 media, fixed and stained for Vps35 (green) and C-terminal APP antibody (APP-CT), which detects full-length-APP and the membrane-bound C-terminal APP fragments. Filipin was used for the staining of free cholesterol. Cells were analyzed by confocal microscopy, and fluorescent signals were quantified using ImageJ software. (**B**) Colocalization of Vps35 and APP-CT was compared between CHOwt and *NPC1*-null cells, *** *p* < 0.001. The data are shown as mean ± SD normalized to control, *n* = 30 cells per group from three different experiments. Scale bar—20 µm (unzoomed figures) and 10 µm (zoomed figures).

**Figure 4 ijms-22-13256-f004:**
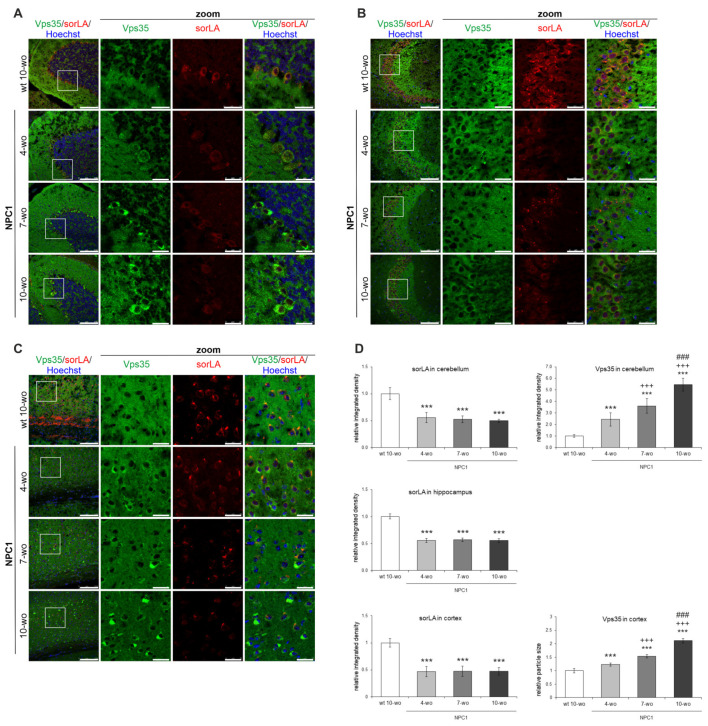
Altered distribution of Vps35 and sorLA in NPC1 mouse brains is already observed at the presymptomatic stage of the disease. Shown are the results of Vps35 and sorLA immunostaining of 4, 7 and 10-week old NPC1 vs. wt mouse cerebella (**A**), hippocampi (**B**) and cortices (**C**). (**D**) The obtained immunofluorescent signals in (**A**–**C**) were quantified using the ImageJ software. The *** (*p* < 0.001) represents statistically significant signal intensities in comparison to wt brains; +++ (*p* < 0.001) in comparison to NPC1 4-week old brains and ### (*p* < 0.001) in comparison to NPC1 7-week old brains. The data are shown as mean ± SD normalized to control and represent data from three different experiments, *n* = 9 slices per animal, 6 animals per group. Scale bar—100 µm (unzoomed pictures) and 20 µm (zoomed pictures).

**Figure 5 ijms-22-13256-f005:**
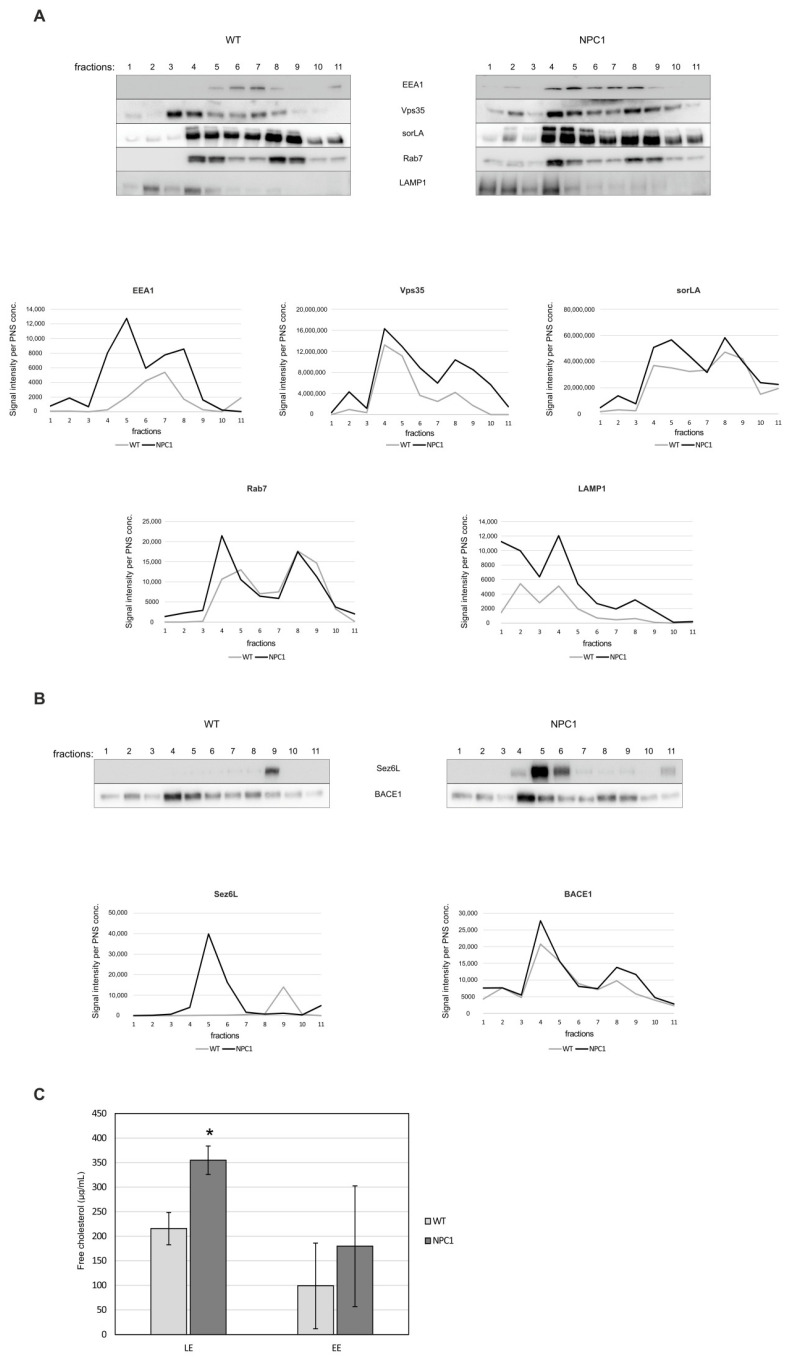
Endosomal fractionation reveals an altered retromer and endocytic distribution between NPC1 and wt-mouse cerebella. Eleven cerebellar fractions (of 1 mL) of 10-week old wt and NPC1 mice were collected from the top of the gradient and were analyzed by Western blotting, as indicated in the material and methods section. (**A**) Western blot analyses of Vps35 (retromer complex protein), sorLA (sortilin-related receptor) and endocytic markers EEA1 (early endosomes), Rab7 (late endosomes) and LAMP1 (lysosomes). Graphs represent quantified protein signals of Vps35, sorLA, EEA1, Rab7 and LAMP1 in each fraction of wt and NPC1 mice, obtained by signal quantification using the ImageJ software, normalized to protein concentration in the corresponding PNS. (**B**) Western blot analyses of BACE1 substrate Sez6L and BACE1. Graphs represent quantified protein signals of Sez6L and BACE1 in each fraction of wt and NPC1 mice, obtained by signal quantification using the ImageJ software, normalized to protein concentration in the corresponding PNS. (**C**) The levels of free cholesterol were determined in early endosome (EE) and late endosome fractions (LE), fractions 4 and 5, and fractions 1 and 2, respectively, from the cerebellum of 10-week old NPC1 and wt-mice by Amplex Red cholesterol assay (Student’s *t*-test, * *p* < 0.05). The shown data are from three different experiments, *n* = 3 cerebella per group.

**Figure 6 ijms-22-13256-f006:**
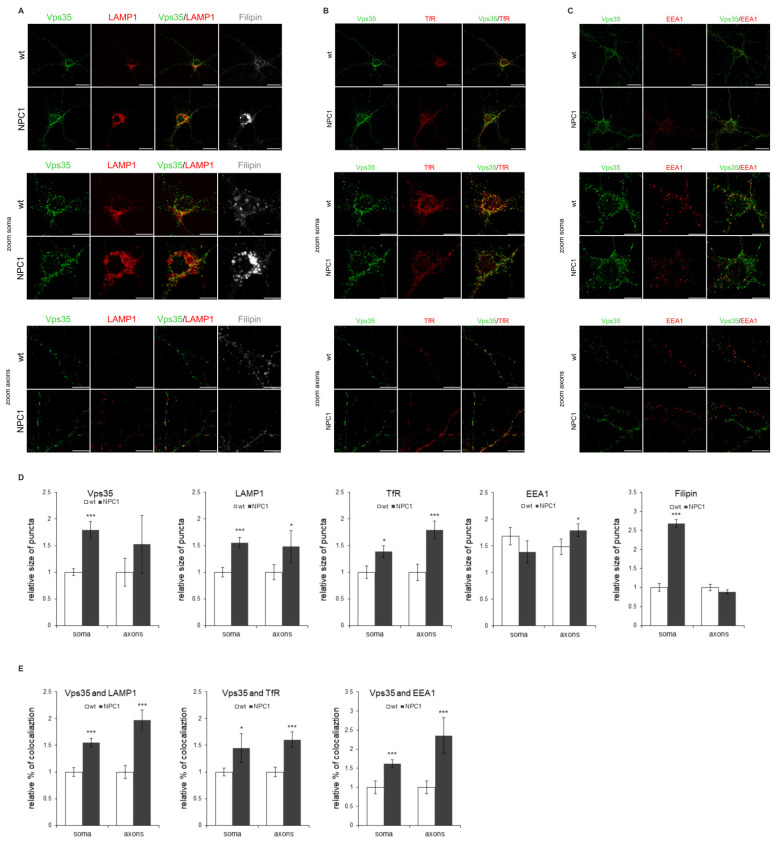
Retromer protein Vps35 accumulates within enlarged endolysosomal EEA1, TfR and LAMP1-positive vesicles in NPC1 hippocampal neurons. Hippocampal neurons were isolated at postnatal day 0 (P0), grown in culture for 14 DIV, fixed and stained for Vps35 (green) and (**A**) lysosomal marker LAMP1 (red), (**B**) transferrin receptor TfR (red) and (**C**) early endosome marker EEA1 (red). Filipin was used for the staining of free cholesterol. Cells were analyzed using confocal microscopy, and fluorescent signals were quantified using ImageJ software. (**D**) Size of Vps35, LAMP1, TfR, EEA1 and filipin-positive vesicles and (**E**) colocalization of Vps35 and EEA1, TfR and LAMP1 in neuronal soma and axons was compared between wt and NPC1 neurons, * *p* < 0.05, *** *p* < 0.001. The data are shown as mean ± SD normalized to control, *n* = 20 neurons per group from three different experiments. Scale bar—20 µm (unzoomed figures) and 10 µm (zoomed figures).

**Figure 7 ijms-22-13256-f007:**
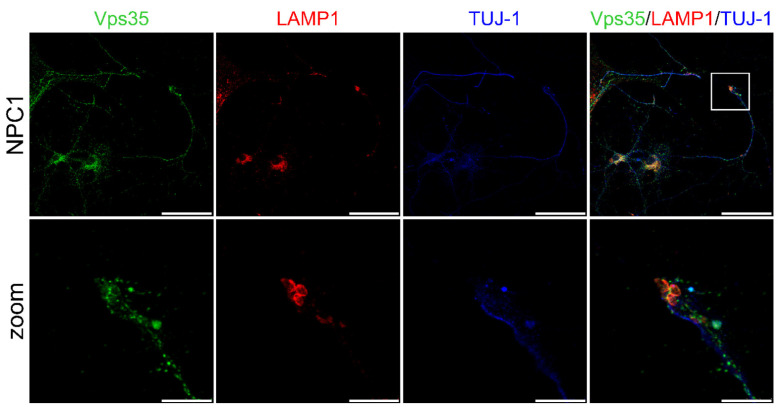
Vps35 is present in axonal swellings of hippocampal NPC1 neurons. Hippocampal neurons were isolated at postnatal day 0 (P0), grown in culture for 14 DIV, fixed and stained for Vps35 (green), lysosomal marker LAMP1 (red) and neuronal marker TUJ-1 (blue). Cells were analyzed using confocal microscopy. The axonal swelling is depicted by white square and in zoomed picture. Scale bar—50 µm (unzoomed) and 10 µm (zoom).

**Figure 8 ijms-22-13256-f008:**
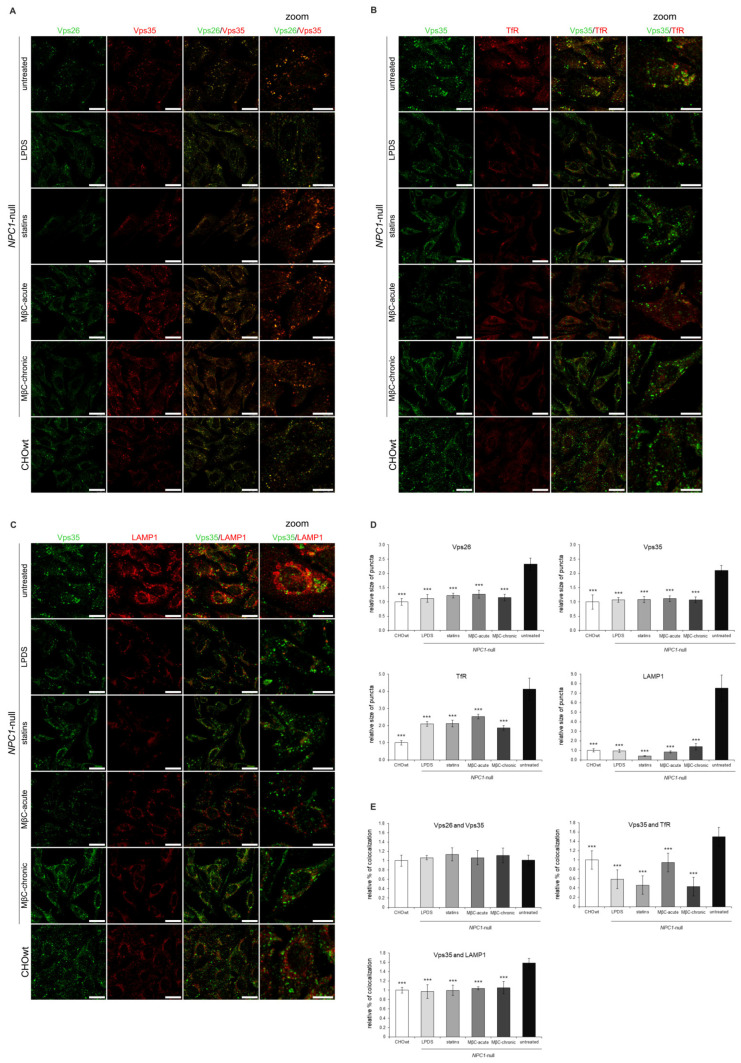
Retromer and endolysosomal trafficking defect is rescued upon cholesterol depletion in vitro in *NPC1*-null cells. Cholesterol levels in *NPC1*-null cells were lowered using four different approaches: LPDS-, LPDS+lovastatin, methyl-β cyclodextrin (MβC)-acute and MβC-chronic treatment. After treatments, cells were fixed and stained for (**A**) Vps26 (green) and Vps35 (red), (**B**) Vps35 (green) and transferrin receptor, TfR (red) and (**C**) Vps35 (green) and lysosomal marker, LAMP1 (red). Cells were analyzed by confocal microscopy, and fluorescent signals were quantified using ImageJ software (**D**,**E**). (**D**) Size of Vps26, Vps35, TfR and LAMP1-positive puncta were compared to the data obtained in untreated *NPC1*-null cells (*** *p* < 0.001). (**E**) Colocalization of the indicated markers was compared to the data obtained in untreated *NPC1*-null cells (*** *p* < 0.001). The data are shown as mean ± SD normalized to control and represent data from three independent experiments, *n* = 30 cells per group. Scale bar—20 µm (unzoomed pictures) and 10 µm (zoomed pictures).

**Figure 9 ijms-22-13256-f009:**
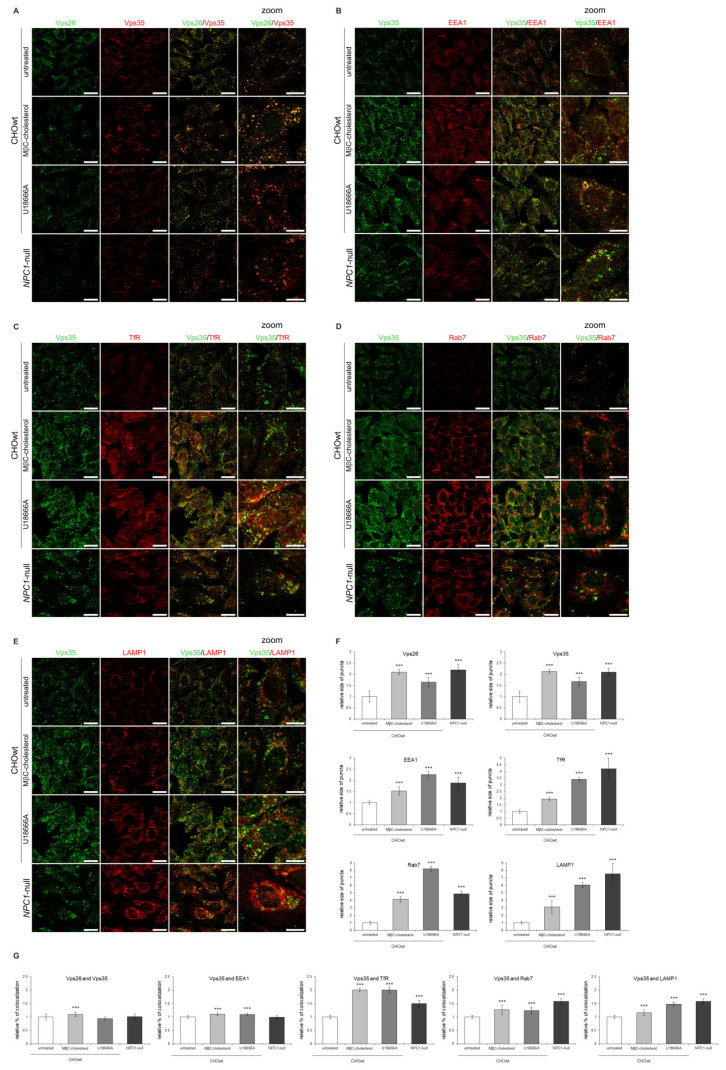
Cholesterol-loading in CHOwt cells causes retromer trafficking defect similar to that as in *NPC1*-null cells. The cells were grown in 10% FBS in DMEM/F12 media. Cholesterol levels were increased in vitro in CHOwt cells using U18666A and MβC-cholesterol complex treatment, after which they were fixed and stained for (**A**) Vps26 (green) and Vps35 (red), or Vps35 (green) and (**B**) early endosome marker EEA1, (**C**) transferrin receptor (TfR, red), (**D**) late endosome marker Rab7 or (**E**) lysosomal marker LAMP1 (red). Cells were analyzed by confocal microscopy, and fluorescent signals were quantified using ImageJ software (**F**,**G**). (**F**) Size of Vps26, Vps35, EEA1, TfR, Rab7 and LAMP1-positive puncta were compared to the data obtained in untreated CHOwt cells (*** *p* < 0.001). (**G**) Colocalization of the indicated markers was compared to the data obtained in untreated CHOwt cells (*** *p* < 0.001). The data are shown as mean ± SD normalized to control and represent data from three independent experiments, *n* = 30 cells per group. Scale bar—20 µm (unzoomed pictures) and 10 µm (zoomed pictures).

**Figure 10 ijms-22-13256-f010:**
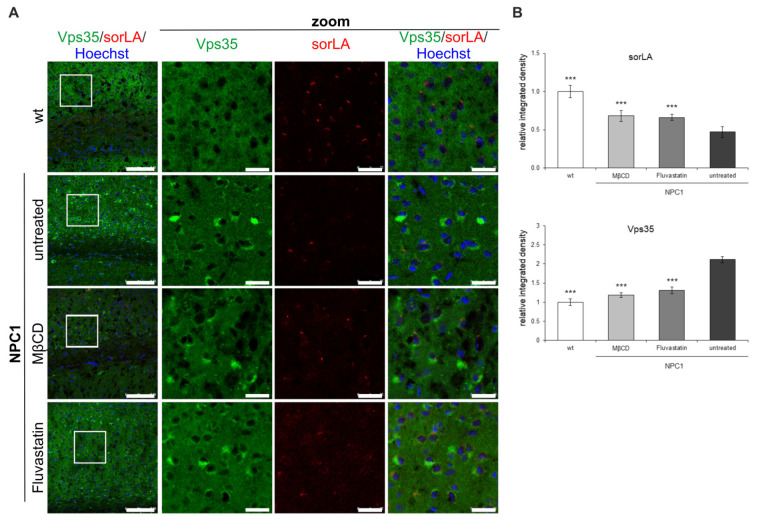
SorLA and Vps35 distribution in the cortex of 10-week old NPC1 mice is rescued by cholesterol depletion in vivo. NPC1 mice were treated with cholesterol-lowering therapy, MβCD or statin (fluvastatin). (**A**) Immunostaining of Vps35 (green) and sorLA (red) was analyzed by confocal microscopy. (**B**) Fluorescent signals (integrated density) were quantified using ImageJ software and compared to untreated NPC1-mice (*** *p* < 0.001). The data are shown as mean ± SD normalized to control and represent data from three different experiments, *n* = 9 slices per animal, 6 animals per group. Scale bar—100 µm (unzoomed pictures) and 20 µm (zoomed pictures).

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
