# Peer review of "Impaired Retromer Function in Niemann-Pick Type C Disease Is Dependent on Intracellular Cholesterol Accumulation"

_ijms, 2021, doi:10.3390/ijms222413256_

Round 1

Reviewer 1 Report

The authors studied a very interesting topic, namely the possible involvement of the retromer complex in a rare lysosomal disorder named Niemann-Pick type C. This could be very important, as components of the complex may represent new drug targets for a disease that imposes a wide range of neurologic symptoms and a reduced life-span of patients. The authors use a mix of cellular and animal models of the disease, and present some results. Unfortunately, the ms shows severe deficits (complete lack of quantitative data for immunostaining; duplicate presentation of micrographs), which dampen interest and clearly preclude publication. The authors should consider the following points:

- Pg. 1 Abstract: The authors should state which strain of NPC1-deficient mice was used.

- Pg. 2, para 2: The role of the SORL1 gene with respect to retromer function should be better explained.

- Pg. 2, para 3: The expression "many pathological features of the most common and complex Alzheimer's disease" looks a bit odd.

- Pg. 2, para 3: The word "insolting" is unclear.

- Pg. 3, para 1; pg. 18: The authors should modify some statements that appear incorrect/imprecise. The in vivo studies in mice used beta-cyclodextrin with hydroxy-propyl residues NOT methyl. Moreover, as the authors are well aware, the "toxicity" depends in vitro on the type of cell and the dose and in vivo on the type of administration (and thus the dose). Distinct clinical trials were performed with intravenous injections and with lumbal punction with completely different dose ranges. Probably the most severe adverse event is futher impairment of hearing (which is already impaired in untreated patients). The authors are strongly advised to describe these issues properly.

- Pgs. 3-18, Figs. 2-4, 6-10, Figs. S1-S4, S6: The observations obtained with immunocyto- and immunohistochemical staining are illustrated solely by (representative?) micrographs with unfounded statements of "significant changes" (e.g. on pg. 8, in legend Fig. 4). The authors must provide quantitative data on the levels (number of labeled puncta etc.) and colocalization of stained proteins to indicate the relevance and soundness of their observations. Their absence precludes publication.

- Pgs. 3-18: All statements with respect to retromer function (e.g. Pg. 5, para 1: "in the enlarged dysfunctional EEA1-positive (Figure 2A)" and "leading to retromer dysfunction") are not supported by data, as the authors have not performed any experiments probing retromer function.

- Pg. 3, para 4: The term "alveolar-shaped" appears a bit misleading.

- Pg. 5 Fig. 2: ATTENTION: panels B and C show the same micrographs, which is technically impossible, if they originate from staining with distinct antibodies. This MUST be corrected! In the absence of appropriate micrographs, the corresponding section of the Results cannot be reviewed or evaluated.

- Pgs. 5-6 Figs. 2 and 3: ATTENTION: panel A of Fig. 2 shows the same micrographs as Fig. 3. Again, this MUST be corrected! In the absence of appropriate micrographs, the corresponding section of the Results cannot be reviewed or evaluated.

- Pg. 8, Fig. 4: The authors should shorten the figure legend.

- Pg. 11, Fig. 5: The authors mention a loading control for Western blots in the Methods section (pg. 23, para 2). However, the quantitative data presented in Fig. 5 are non-normalized values, which is not sufficient to judge the distribution of the proteins. Moreover, it is unclear what the lanes labeled 1-11 represent. This should be explained properly.

- Pg. 13, Fig. 7: The legends indicates staining of filipin, but none is shown in the micrographs. This must be corrected.

- Pg. 16, para 1: The authors' statements imply that treatment of cells with MbetaC complexed to cholesterol or with U18666A are equivalent in inducing "cholesterol loading". This is not the case, as each treatment has a distinct effect. The rationale for these experiments is unclear and thus must be explained better.

- Pg. 16, Figure 8: ATTENTION: panel C and E show the same micrograph at least for Vps35, which is technically impossible. Again, this MUST be corrected! In the absence of appropriate micrographs, the corresponding section of the Results cannot be reviewed or evaluated.

- Pg. 17, Figure 9: ATTENTION: panel C and E show the same micrograph, which is also presented in Fig. 2 and 3. This MUST be corrected! In the absence of appropriate micrographs, the corresponding section of the Results cannot be reviewed.

- All figures with micrographs: The authors must add scale bars.

- Supplementary figures: It is unclear, why data that appear as important, are shown as supplementary figures. The authors should consider to integrate the data in the main figures.

- Throughout the ms, English language should be checked.

Author Response

Special Issue "Advances in Knowledge in Niemann-Pick Disease Type C: Facts and Perspectives- 2nd Edition"

IJMS Editorial Office

November 19th, 2021.

Dear Editors,

We very much appreciate the review of our manuscript submitted to the Special Issue "Advances in Knowledge in Niemann-Pick Disease Type C: Facts and Perspectives- 2nd Edition" of the International Journal of Molecular Sciences (IJMS-1472106) entitled “Impaired retromer function in Niemann-Pick type C disease is dependent on intracellular cholesterol accumulation”.

We have carefully considered all the comments raised by the reviewers and have addressed them all in the revised version of our manuscript. Please below find our point-by-point response to the comments brought up by the reviewers. The page numbers that we refer to in our response are included in the revised version of the manuscript with track changes. Also, we have carefully considered the proposed English language editing which was performed by a native English-speaking colleague.

We very much appreciate positive comments from reviewers and are thankful for their suggestions, which have certainly contributed to the improvement of our manuscript. We hope that our manuscript in the revised form will be acceptable for publication in International Journal of Molecular Sciences.

Sincerely,

Silva Hecimovic, PhD

Head, Laboratory for Neurodegenerative Disease Research

Division of Molecular Medicine

Rudjer Boskovic Institute

Bijenicka c. 54, 10000 Zagreb - Croatia

Tel/Fax: +385-1-4571327

email: silva.hecimovic@irb.hr

 Reviewers’ Comments and Suggestions for Authors

 Reviewer 1:

The authors studied a very interesting topic, namely the possible involvement of the retromer complex in a rare lysosomal disorder named Niemann-Pick type C. This could be very important, as components of the complex may represent new drug targets for a disease that imposes a wide range of neurologic symptoms and a reduced life-span of patients. The authors use a mix of cellular and animal models of the disease, and present some results. Unfortunately, the ms shows severe deficits (complete lack of quantitative data for immunostaining; duplicate presentation of micrographs), which dampen interest and clearly preclude publication. The authors should consider the following points:

- Pg. 1 Abstract: The authors should state which strain of NPC1-deficient mice was used.

Our response: We thank the reviewer for noting the lack of information on NPC1 mouse strain in the Abstract. In the revised manuscript, in the Abstract (pg.1) we have included the mouse strain details:

“(BALB/cNctr-Npc1m1N)“.

- Pg. 2, para 2: The role of the SORL1 gene with respect to retromer function should be better explained.

Our response: We very much appreciate the reviewer’s comment. In the revised manuscript (pg.1 and 2) we have addressed the role of sorLA in retromer function:

“This trimer directly recognizes and binds to the cargo molecules (kinases, phosphatases and signaling receptors) forming a cargo-recognition complex. Intracellular sorting receptors, such as sorting-related receptor with A-type repeats (sorLA- also known as SORL1 and LR11), acts as a bridge between the retromer and the cargo proteins, and regulates their correct location and function within the cell. SorLA is expressed mainly in neurons in the CNS and is involved in recognition of multiple cargo molecules and their retrograde trafficking from early endosomes to the trans-Golgi network (TGN) via the retromer complex. Given that the retromer complex plays a critical role in endolysosomal trafficking and function, it is reasonable to suggest that retromer and/or sorting receptor defect may be a central hub contributing to neurodegeneration.“

- Pg. 2, para 3: The expression "many pathological features of the most common and complex Alzheimer's disease" looks a bit odd.

Our response: According to the reviewer’s comment, we have changed the wording and have included “several...” instead of many pathological features...“ (pg. 2, para 3). We would like to note that numerous shared pathological features were indicated in submitted as well as in the revised version of the manuscript:

“It is intriguing that Alzheimer's disease (AD) and a rare inherited lipid storage disorder Niemann-Pick type C (NPC) share several pathological features, including dysfunctional endolysosomal system, enhanced amyloidogenic APP processing (resulting in accumulation of the C-terminal APP fragments (CTFβ/C99) and Aβ peptides), tau pathology, apolipoprotein E ε4 risk factor, activation of astrocytes and microglia (neuroinflammation), synaptic dysfunction and neurodegeneration [32,33].“

- Pg. 2, para 3: The word "insolting" is unclear.

Our response: We very much appreciate the reviewer’s comment and apologize for the wrong spelling of the word. “insulting” that was used to address the lipid species that is primarily affected  by NPC1/NPC2 dysfunction and is driving the pathological processes in NPC. In the revised manuscript we used instead the word “offending lipid” which has been used previously in the literature in the context of NPC disease (pg. 3, para 1; pg. 21).

“Therefore, one needs to identify an offending lipid in NPC, that initiates pathological processes, correlate it with a particular functional defect in order to employ an effective, lipid-lowering strategy against NPC.“

- Pg. 3, para 1; pg. 18: The authors should modify some statements that appear incorrect/imprecise. The in vivo studies in mice used beta-cyclodextrin with hydroxy-propyl residues NOT methyl. Moreover, as the authors are well aware, the "toxicity" depends in vitro on the type of cell and the dose and in vivo on the type of administration (and thus the dose). Distinct clinical trials were performed with intravenous injections and with lumbal punction with completely different dose ranges. Probably the most severe adverse event is futher impairment of hearing (which is already impaired in untreated patients). The authors are strongly advised to describe these issues properly.

Our response: We are grateful to the reviewer for this comment. In the revised manuscript we have addressed the cyclodextrin-based treatments in preclinical and clinical studies in NPC animal models (mice and cats) and in NPC patients in more detail. We have also indicated correctly the use of 2-hydroxypropyl-ß-cyclodextrin instead of methyl- ß-cyclodextrin in these studies (pg. 3, para 1 and pg. 28).

“This has been supported by the preclinical and clinical trials with a cholesterol extracting compound cyclodextrin (2-hydroxypropyl-β-cyclodextrin, HPβCD) [39–43]. Indeed, administration of HPβCD, both systematically and directly into the CNS, has shown the greatest disease amelioration in NPC1 mouse and feline models, including rescue of lethality and improved pathological hallmarks of NPC disease [39,40]. However, ototoxicity, e.g. further impairment of hearing which is already present in NPC, was detected as the most severe adverse event. Importantly, direct intracerebroventricular (ICV) injection overcame HPβCD inability to efficiently cross the blood–brain barrier and resulted in a complete prevention of neurodegeneration of Purkinje cells in NPC1 mice. Clinical trials with intrathecal (IT) HPβCD administration showed both biomarker and clinical evidence of efficacy in patients with NPC disease [41,42]. Although the doses applied were generally well tolerated, dose-limiting side-effects including the transient post-dose ataxia and fatigue were reported, in addition to ototoxicity. Recent studies on intravenous (IV) administration of HPβCD to moderately and severely affected NPC patients showed neurologic and neurocognitive benefits in most patients with IV application alone, independent of the IT administration when combined with IV [43]. No unexpected safety issues were experienced and the systemic reactions that were noted did not lead to treatment discontinuation. Notably, hearing loss was not reported in the patients receiving IV therapy, as has been reported in previous clinical trials on NPC patients with IT administration [41-43]. The rate of disease progression appeared to be less than expected in some individuals receiving IV HPβCD. Thus, HPβCD treatment seems to play a significant role in targeting the cellular cholesterol burden known to be present in NPC, but optimal dosing, dosing interval and route that would enhance outcome for patients with NPC are still to be determined. It is likely that a multi-system disease such as NPC would require a multitargeted treatment approach. In addition, the heterogeneity of the disease is important to consider for treatment strategies against NPC, as an approach in one patient may not be the most appropriate for another.“

“Indeed, it has been previously shown that cholesterol depletion using cyclodextrins (HPβCD) rescues lethality of NPC1 mice and improves pathological hallmarks of NPC disease [39].”

- Pgs. 3-18, Figs. 2-4, 6-10, Figs. S1-S4, S6: The observations obtained with immunocyto- and immunohistochemical staining are illustrated solely by (representative?) micrographs with unfounded statements of "significant changes" (e.g. on pg. 8, in legend Fig. 4). The authors must provide quantitative data on the levels (number of labeled puncta etc.) and colocalization of stained proteins to indicate the relevance and soundness of their observations. Their absence precludes publication.

Our response: We are very much grateful to the reviewer comment and agree that the first version of our manuscript had rather subjective interpretation of the data obtained by immunofluorescent staining. In the revised manuscript we have addressed this criticism and have included quantifications of all immunocytochemistry and immunohistochemistry fluorescent images. As suggested we included quantification of the size and/or number of the puncta, % of colocalization and integrated density signal when appropriate. In the Material and methods section of the revised manuscript we included the information on the ImageJ software that we used to analyze this (pg. 34 and 35). These include changes in Figures 1-4 and 6-10, except Figure 7. In Figure 7 we aimed to describe changes in axonal swelling only in NPC1 hippocampal neurons qualitatively. In addition, we quantitatively analyzed immunofluorescent signals in supplementary figures Figure S1-S4 and S6, except Figure S3 that already contains quantitative analysis of western blot signals.     

- Pgs. 3-18: All statements with respect to retromer function (e.g. Pg. 5, para 1: "in the enlarged dysfunctional EEA1-positive (Figure 2A)" and "leading to retromer dysfunction") are not supported by data, as the authors have not performed any experiments probing retromer function.

Our response: We very much appreciate the reviewer comment and agree that the retromer function was not examined directly in our study. In the revised version of the manuscript throughout the whole text (pg. 5-32), including the text related to the results in Figure 2A (pg. 7, para 1), we have indicated the specific impairments in retromer trafficking/distribution and/or transport that we have observed in multiple in vitro and in vivo NPC models.

- Pg. 3, para 4: The term "alveolar-shaped" appears a bit misleading.

Our response: We appreciate the reviewer’s suggestion about the used term “alveolar-shaped”. We would like to note that the used term “alveolar-shaped vesicles” was introduced in the manuscript Marquer et al. Nat Commun 2016;7:11919. doi: 10.1038/ncomms11919 (https://pubmed.ncbi.nlm.nih.gov/27336679/)  which described similar retromer accumulation and impaired transport in Arf6 KO cells. However, we acknowledge that this term could be misleading. Therefore, throughout the manuscriptwe used “retromer accumulation within enlarged vesicles”, or similar, instead of “alveolar-shaped vesicles”.

- Pg. 5 Fig. 2: ATTENTION: panels B and C show the same micrographs, which is technically impossible, if they originate from staining with distinct antibodies. This MUST be corrected! In the absence of appropriate micrographs, the corresponding section of the Results cannot be reviewed or evaluated.

Our response: We thank the reviewer for this comment. However, we do not agree since it is technically possible to perform staining using several antibodies on same immunocytochemistry sample if the primary antibodies originate from different species that can be further labelled with different fluorescent-labelled secondary antibodies, as was indicated in Materials and methods section (pg. 34, para 4.6. Immunocytochemistry and confocal microscopy). For example, Figure 2B and C shows same cells that were stained with primary anti-Vps35 (goat) and secondary donkey anti-goate-Alexa Fluor 647, primary anti-TfR (mouse) and secondary donkey anti-mouse-Alexa Fluor 594 and primary anti-LAMP1 (rabbit) and secondary donkey anti-rabbit-Alexa Fluor 488, and filipin (UV). Thus, the result shown in Figures 2B and C could have been merged in one figure using three different colours. However, under these setting the analysis of the figures, the visualisation and colocalization analysis would be very complex and hard to follow. Therefore, to simplify, we showed these stainings using only two colours (red/green images) to demonstrate colocalization of the two markers as well as the size of the puncta of the stained vesicles.  Figure 7, which analysed axonal swelling, contains staining of the three markers using three different primary and secondary antibodies, thus, showing the technical possibility of such analysis.

- Pgs. 5-6 Figs. 2 and 3: ATTENTION: panel A of Fig. 2 shows the same micrographs as Fig. 3. Again, this MUST be corrected! In the absence of appropriate micrographs, the corresponding section of the Results cannot be reviewed or evaluated.

Our response: Please find our response to the comment above concerning the performed immunostaining in Figure 2.

- Pg. 8, Fig. 4: The authors should shorten the figure legend.

Our response: In the revised manuscript we have shortened the figure legend of Figure 4.

- Pg. 11, Fig. 5: The authors mention a loading control for Western blots in the Methods section (pg. 23, para 2). However, the quantitative data presented in Fig. 5 are non-normalized values, which is not sufficient to judge the distribution of the proteins. Moreover, it is unclear what the lanes labeled 1-11 represent. This should be explained properly.

Our response: We thank the reviewer for the comment as the western blot analysis of endosome fractions was not described properly and in sufficient detail. In the revised manuscript, both in the Material and methods section (pg. 34 and 35) and in the figure legend of Figure 5 (pg. 18) we have addressed this criticism.

4.5. Western blot

...„ To analyze the isolated endosome fractions by Western blotting the samples of the wt and NPC1 mouse brains were normalized prior ultracentrifugation and endosome fractionation by loading the same amount of the postnuclear supernatant (PNS) into the gradient. Upon ultracentifugation, equal volumes of isolated fractions were loaded and alayzed by western blotting. Protein signals were quantified in each fraction using the ImageJ software. The obtained signal intensity represents the amount of the analyzed vesicular marker in each fraction.“

4.9. Endosome fractionation

...“The equal amount of PNS of wt and NPC1-mouse brains were loaded at the bottom of an ultracentrifuge tube (14×89 mm, Beckman) and adjusted to 40.6% sucrose in 3 mM imidazole (Fluka), pH 7.4 and 1 mM EDTA (Kemika) by mixing with 62% sucrose in 3 mM imidazole (pH 7.4) and 1 mM EDTA. Three milliliters of PNS-mixture were overlaid with 4.5 mL of 35%, 3 mL of 25% and 1 mL of 8% sucrose in 3 mM imidazole (pH 7.4) and 1 mM EDTA.“.

„Figure 5. Endosomal fractionation reveals altered retromer and endocytic dystribution between NPC1- and wt-mouse cerebella. Eleven cerebellar fractions (of 1 ml) of 10-weeks old wt and NPC1 mice were collected from the top of the gradient and were analyzed by Western blotting as indicated in Material and methods section. A) Western blot analyses of Vps35 (retromer complex protein), sorLA (sortilin-related receptor) and endocytic markers EEA1 (early endosomes), Rab7 (late endosomes) and LAMP1 (lysosomes). Graphs represent quantified protein signals of Vps35, sorLA, EEA1, Rab7, LAMP1 in each fraction of wt and NPC1 mice, obtained by signal quantification using the ImageJ software. B) Western blot analyses of BACE1 substrate Sez6L and BACE1. Graphs represent quantified protein signals of Sez6L and BACE1 in each fraction of wt and NPC1 mice, obtained by signal quantification using the ImageJ software. C) The levels of free cholesterol were determined in early endosome (EE) and late endosome fractions (LE) from the cerebellum of 10-weeks old NPC1- and wt-mice by Amplex Red cholesterol assay.”

- Pg. 13, Fig. 7: The legends indicates staining of filipin, but none is shown in the micrographs. This must be corrected.

Our response: We apologize for the carelessness in preparing the legend of Figure 7. In the revised manuscript, the mentioned filipin staining in the legend of Figure 7 was omitted.

- Pg. 16, para 1: The authors' statements imply that treatment of cells with MbetaC complexed to cholesterol or with U18666A are equivalent in inducing "cholesterol loading". This is not the case, as each treatment has a distinct effect. The rationale for these experiments is unclear and thus must be explained better.

Our response: We very much appreciate the reviewer’s comment and agree that the cholesterol-loading approaches were not decribed in sufficient detail. In the revised manuscript we included a detailed explanation of the two approaches used for cholesterol-loading (pg. 25). We would like to note that we have also added a reference of our previous manuscript in which we have utilized the same treatment approaches for cholesterol-loading in CHO cells (Malnar et al. Biochim Biophys Acta 2012, https://doi.org/10.1016/j.bbadis.2012.04.002.).  

“Secondly, to load cholesterol in vitro in CHOwt cells we used two different approaches: U18666A-treatment, that mimics intracellular cholesterol acccumulation within late endosomes/lysosomes as in NPC1-null cells, and MβC-cholesterol complex treatment, that causes increased cholesterol in all cellular membranes (plasma membrane and intracellular membranes) [49]. These cholesterol-loading approaches in CHOwt cells caused increased total and free cholesterol levels (Figure S6A), accumulation of unesterified cholesterol in late endosomes and lysosomes (U-treatment) and enhanced cholesterol staining within the cell (MβC-treatment) (Figure S6B) as well as lysosomal impairment (Figure S6C).“

- Pg. 16, Figure 8: ATTENTION: panel C and E show the same micrograph at least for Vps35, which is technically impossible. Again, this MUST be corrected! In the absence of appropriate micrographs, the corresponding section of the Results cannot be reviewed or evaluated.

Our response: Please find our response to the comment above concerning the performed immunostaining in Figure 2.

- Pg. 17, Figure 9: ATTENTION: panel C and E show the same micrograph, which is also presented in Fig. 2 and 3. This MUST be corrected! In the absence of appropriate micrographs, the corresponding section of the Results cannot be reviewed.

Our response: Please find our response to the comment above concerning the performed immunostaining in Figure 2.

- All figures with micrographs: The authors must add scale bars.

Our response: We apologize for the lack of the scale bars in our images. In the revised manuscript we have included scales of all images.

- Supplementary figures: It is unclear, why data that appear as important, are shown as supplementary figures. The authors should consider to integrate the data in the main figures.

Our response: We very much appreciate the reviewer’s suggestion to include the data from the Supplementary figures into the main article. However, due to already high number of figures and the data presented in them, we have not included any new figures in the main text of the revised manuscript. However, we kindly appreciate if the reviewer could suggest which supplementary figure/data she/he would like to add to the main text of the revised manuscript, if she/he still finds it necessary.

- Throughout the ms, English language should be checked.

Our response: We very much thank the reviewer for the suggestion to improve the English language in the manuscript. The English language was indeed edited by a native English-speaking colleague.

Reviewer 2 Report

The paper from Dominko and co-workers describes a retromer dysfunction in different models (both in vitro and in vivo) of Niemann Pick type C disease (NPC). The authors found an altered retromer distribution in NPC models and demonstrated that the depletion of cholesterol both in vitro and in vivo was able to rescue retromer alterations. The paper is well-written and is very interesting but it requires a major revision for the following reasons:

  • the figures are described in a qualitative way but in some of them “a significant” decrease in the expression of some markers is described (e.g in Fig. 4); thus, immunofluorescence images aimed at describing variation in the expression of retromer markers should be quantitively analyzed and the number of replicates should be indicated;
  • the number of replicates used for the histograms on figures 1, 5 and in the supplementary should be indicated;
  • the percentage of colocalization should be calculated for immunofluorescence images.

Minor points:

  • On page 2 few words to better delineate the role played by SORL1 in retromer function should be added
  • please, add the complete form “trans Golgi Network” before its abbreviations TGN on page 2
  • the same for APP and fl-APP on pages 2 and 6, respectively

figures 6 and 8 should be substituted with higher-definition images. 

Author Response

Special Issue "Advances in Knowledge in Niemann-Pick Disease Type C: Facts and Perspectives- 2nd Edition"

IJMS Editorial Office

November 19th, 2021.

Dear Editors,

We very much appreciate the review of our manuscript submitted to the Special Issue "Advances in Knowledge in Niemann-Pick Disease Type C: Facts and Perspectives- 2nd Edition" of the International Journal of Molecular Sciences (IJMS-1472106) entitled “Impaired retromer function in Niemann-Pick type C disease is dependent on intracellular cholesterol accumulation”.

We have carefully considered all the comments raised by the reviewers and have addressed them all in the revised version of our manuscript. Please below find our point-by-point response to the comments brought up by the reviewers. The page numbers that we refer to in our response are included in the revised version of the manuscript with track changes. Also, we have carefully considered the proposed English language editing which was performed by a native English-speaking colleague.

We very much appreciate positive comments from reviewers and are thankful for their suggestions, which have certainly contributed to the improvement of our manuscript. We hope that our manuscript in the revised form will be acceptable for publication in International Journal of Molecular Sciences.

Sincerely,

Silva Hecimovic, PhD

Head, Laboratory for Neurodegenerative Disease Research

Division of Molecular Medicine

Rudjer Boskovic Institute

Bijenicka c. 54, 10000 Zagreb - Croatia

Tel/Fax: +385-1-4571327

email: silva.hecimovic@irb.hr

Reviewer 2:

Comments and Suggestions for Authors

The paper from Dominko and co-workers describes a retromer dysfunction in different models (both in vitro and in vivo) of Niemann Pick type C disease (NPC). The authors found an altered retromer distribution in NPC models and demonstrated that the depletion of cholesterol both in vitro and in vivo was able to rescue retromer alterations. The paper is well-written and is very interesting but it requires a major revision for the following reasons:

  • the figures are described in a qualitative way but in some of them “a significant” decrease in the expression of some markers is described (e.g in Fig. 4); thus, immunofluorescence images aimed at describing variation in the expression of retromer markers should be quantitively analyzed and the number of replicates should be indicated;

Our response: We are very much grateful to the reviewer comment and agree that the first version of our manuscript had rather subjective interpretation of the data obtained by immunofluorescent staining. In the revised manuscript we have addressed this criticism and have included quantifications of all immunocytochemistry and immunohistochemistry fluorescent images. We included quantification of the size and/or number of the puncta, % of colocalization and integrated density signal when appropriate. In the Material and methods section of the revised manuscript we included the information on the ImageJ software that we used to analyze this (pg. 34 and 35). These include changes in Figures 1-4 and 6-10, except Figure 7. In Figure 7 we aimed to describe changes in axonal swelling only in NPC1 hippocampal neurons qualitatively. In addition, we quantitatively analyzed immunofluorescent signals in supplementary figures Figure S1-S4 and S6, except Figure S3 that already contains quantitative analysis of western blot signals.    

  • the number of replicates used for the histograms on figures 1, 5 and in the supplementary should be indicated;

Our response: We are thankful the reviewer for this observation. In the revised manuscript we have added the information on the number of replicates used where appropriate, including figures 1 and 5.

  • the percentage of colocalization should be calculated for immunofluorescence images.

Our response: Please find our response to the comment above concerning the quantitative analysis of immunofluorescence images.

Minor points:

  • On page 2 few words to better delineate the role played by SORL1 in retromer function should be added

Our response: We very much appreciate the reviewer’s comment. In the revised manuscript (pg.1 and 2) we have addressed the role of sorLA in retromer function:

“This trimer directly recognizes and binds to the cargo molecules (kinesis, phosphatases and signaling receptors) forming a cargo-recognition complex. Intracellular sorting receptors, such as sorting-related receptor with A-type repeats (sorLA- also known as SORL1 and LR11), act as a bridge between the retromer and the cargo proteins, and regulate their correct location within the cell and function. SorLA is expressed mainly in neurons in the CNS and is involved in recognition of multiple cargo molecules and their retrograde trafficking from early endosomes to the trans-Golgi network (TGN) via retromer complex. Given that the retromer complex plays a critical role in endolysosomal trafficking and its function, it is reasonable to suggest that retromer and/or sorting receptor defect may be a central hub contributing to neurodegeneration.“

  • please, add the complete form “trans Golgi Network” before its abbreviations TGN on page 2

Our response: We appreciate the reviewer’s comment. In the revised manuscript we included the complete form of “trans Golgi Network” when first mentioned in the text on pg. 2, para 1.

  • the same for APP and fl-APP on pages 2 and 6, respectively

Our response: We appreciate the reviewer’s comment. In the revised manuscript we included the complete form for APP (β-amyloid precursor protein (APP) on pg.2, para 2) and for fl-APP (the full length-APP protein (fl-APP) on pg. 10).

  • figures 6 and 8 should be substituted with higher-definition images.

Our response: We thank the reviewer for the suggestion. The higher-definition images (800 dpi) of Figures 6 and 8 were included in the revised version of the manuscript.

Round 2

Reviewer 1 Report

The authors responded to the comments in a satisfactory manner. The ms is now acceptable for publication.

Reviewer 2 Report

The paper has been extensively improved and all suggestions have been taken into account. Undoubtedly, it is acceptable for the publication.